# Poverty eradication in a carbon constrained world

Klaus Hubacek [1,2], Giovanni Baiocchi[1,3], Kuishuang Feng [1] & Anand Patwardhan[4]

The UN Framework Convention on Climate Change aims to keep warming below 2 °C while recognizing developing countries' right to eradicate extreme poverty. Poverty eradication is also the first of the Sustainable Development Goals. This paper investigates potential consequences for climate targets of achieving poverty eradication. We find that eradicating extreme poverty, i.e., moving people to an income above $1.9 purchasing power parity (PPP) a day, does not jeopardize the climate target even in the absence of climate policies and with current technologies. On the other hand, bringing everybody to a still modest expenditure level of at least $2.97 PPP would have long-term consequences on achieving emission targets. Compared to the reference mitigation pathway, eradicating extreme poverty increases the effort by 2.8% whereas bringing everybody to at least $2.97 PPP would increase the required mitigation rate by 27%. Given that the top 10% global income earners are responsible for 36% of the current carbon footprint of households; the discourse should address income distribution and the carbon intensity of lifestyles.

[1] Department of Geographical Sciences, University of Maryland, College Park, MD 20742, USA. [2] Department of Environmental Studies, Masaryk University, Brno, Czech Republic. [3] Department of Economics, University of Maryland, University of Maryland, College Park, MD 20742, USA. [4] School of Public Policy, University of Maryland, College Park, MD 20742, USA. Correspondence and requests for materials should be addressed to K.H. (email: Hubacek@umd.edu) or to K.F. (email: kfeng@umd.edu)

"End poverty in all its forms everywhere" is the first of the United Nation's Sustainable Development Goals (SDGs) adopted last September; setting targets of eradicating extreme poverty by 2030 for all people everywhere. In parallel another United Nations process took place that culminated in December 2015 where 195 countries adopted the new Paris Agreement under the United Nations Framework Convention on Climate Change aiming at keeping warming to well below 2 °C above pre-industrial levels in the long term while recognizing developing countries right to eradicate extreme poverty and develop sustainably (see UNFCCC[1]). These agreements provide a basis for putting the world economies on a sustainable pathway. However, both agreements do not prescribe how these ambitious goals may be achieved in a compatible manner, nor how the burden or responsibility of achieving them may be shared.

These issues must be considered within the context of global economic inequality and historical responsibility given that developed countries are responsible for the majority of fossil $CO_2$ emissions from 1750 to 2010[2] and the fact that only 18% of the current global population enjoy First World living standards[3] despite the large impacts from current levels of resource extraction and through-put, while at the same time large parts of the global population live at desperate poverty levels. For example, 770 million people lived on <1.90$ a day in purchasing power parities (PPP) (referred to as extreme poverty) in 2013[4], and about half the world population lives on <2.97$ PPP a day[5].

The global community has responded with numerous policy goals to address the issue of extreme poverty, as well as Sustainable Development. Sustainable Development requires to meet the twin objective: to ensure that all people have the resources needed, such as food, water, access to health care and energy, to fulfill their human rights, and to ensure that humanity's use of natural resources does not stress critical earth system processes. According to the report The Action Agenda for Sustainable Development by the UN's Council of the Sustainable Development Solutions Network (or SDSN[6], "sustainable economic growth" should allow "all low-income countries to reach the per capita income threshold of middle-income countries by 2030."

Achieving the Paris target of limiting climate change to well below 2 °C would not only require aggressive decarbonization in rich countries but may also limit the aspirations of poor countries as aggressive limits on greenhouse gas emissions can limit the options for energy sector growth; and growth in energy use is correlated with economic growth and poverty reduction, although the strength of this link is subject to change dependent on technology and consumption patterns[7, 8]. There is a rich literature that views poverty as multidimensional and poorly represented by income measures (e.g., refs. [8–10]). The SDGs themselves make reference to eliminating poverty in all its dimensions and a focus merely on consumption represents a limited point of view but is often seen as a reasonable approach for capturing extreme poverty. There are numerous studies that investigate distributional effects of carbon and energy policies at regional and national levels[11, 12]. But to our knowledge there are currently only very few global quantitative analyses (e.g., ref. [13]) of what poverty alleviation would mean in terms of carbon implications and this is only a first step in this direction that uses detailed information on consumption patterns of people living at the edge of subsistence which is highly relevant to address poverty issues. Despite limitations, quantifying the "climate-development conflict" through greenhouse gas emissions associated with energy consumption[14] is a very useful first approximation as energy is part of many household consumption activities as well as being an essential input to production of goods and services in all stages of global supply chains.

We find that in 2010, the global elite or top 10% of income earners were responsible for 36% of global carbon emissions whereas the extreme poor accounting for 836 million people, that was 12% of the global population, contributed only 4% of global emissions. Overall, the bottom half of global income earners caused only about 13%. Based on these specific carbon footprints for different income categories across the globe we find that lifting people out of extreme poverty has only relative little carbon implications with a projected increase of about 0.05 °C above the IPCC base run by the end of the 21st century. However, when moving the global poor, i.e., everybody below an income of $2.97 (in PPP) to the next income level, which is by the standards of industrialized countries still fairly modest, we would add another 0.6 °C by the end of the century. This more ambitious scenario would significantly increase the required speed and extent of future emission reductions. Given that so far efficiency gains have not been able to keep up with additional emissions a greater focus on demand-side measures jointly with lifestyle and behavioral change is called for especially considering the huge global carbon inequality staying in the way of efforts towards a low carbon society.

## Results

We follow a simplifying approach to energy consumption and energy poverty in households, especially in the Global South, ignoring the role energy plays for households as an important prerequisite to enable capabilities (e.g., refs. [12, 15]). Our research complements recent World Bank reports by Fay et al. and Hallegatte et al.[16, 17], that make the case that climate mitigation policies can be introduced without slowing down poverty reduction, but these do not calculate the impact of doing so on global emissions. We approach this problem in several steps: we first calculate the additional carbon caused by higher levels of income and associated expenditure patterns; we then calculate the temperature increase associated with higher carbon emissions driven by increased consumption and associated increase in production and production capacity. These changes are introduced annually until 2030, in line with the SDG. We keep technology, i.e., carbon intensity constant until 2030, considering the uncertainty in technology, economic growth and policies (see e.g., The International Energy Outlook[18]. Beyond 2030, we do not make any assumptions about technical change but rather ask which annual reduction in carbon emissions is required to compensate for the additional carbon emitted through higher consumption levels (see Rozenberg et al.[19] for a similar question). For calculating carbon footprints for different income groups, we follow the consumption-based approach to carbon accounting based on multi-regional input–output (IO) analysis. This allows us to account for carbon emissions throughout global supply chains, which are then allocated to the final consumer[20]. It not only enables us to account for household's carbon emissions associated with direct emissions associated with heating and cooling, cooking and transport but also accounting for the carbon emitted during the production of products and services consumed by different types of households. For the consumption patterns in developing countries, we use the World Bank's Global Consumption Database[21]. This database contains the most detailed depiction of consumption patterns and lifestyles representing 4.5 billion people. For consumption patterns of household groups in developed countries we use consumer expenditure surveys of national statistical offices for the US[22], the EU[23], Australia[24], and Japan[25].

**Global carbon inequality**. Once we calculate the carbon footprint for each income group (Fig. 1) we get the following picture for

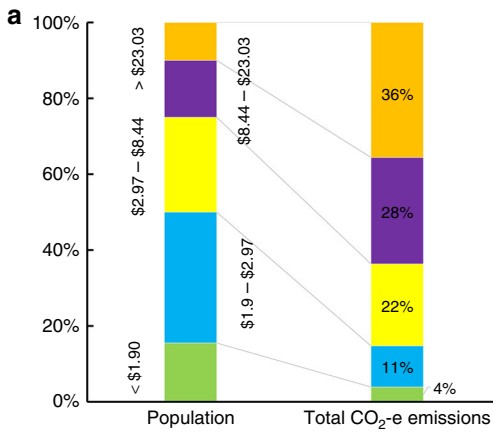
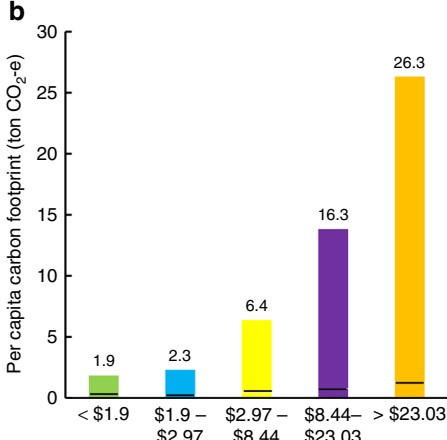

**Fig. 1** Global income and carbon distribution in 2010 for household final demand plus associated government expenditure and capital formation. **a** Global population share and global carbon contribution per expenditure category. Colors represent different expenditure categories and their respective shares of the global population and global carbon emissions. **b** Per Capita carbon footprint per expenditure category in tons of $CO_2$-e. Dotted line separates the direct and indirect emissions for each consumer groups (lower part is the direct emissions and upper part is the indirect or upstream emissions in the supply chain)

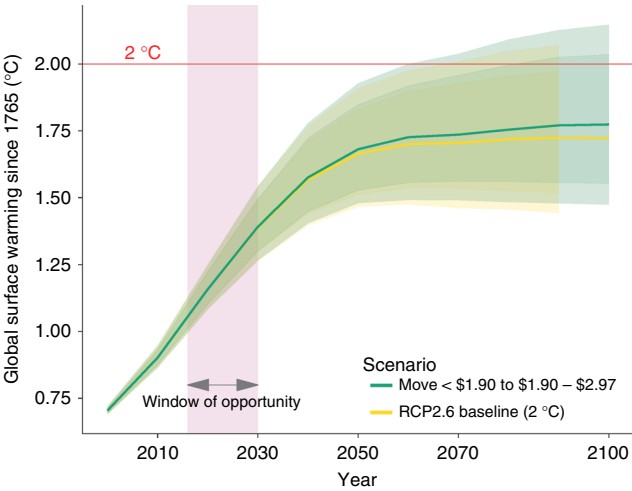

**Fig. 2** Predicted increases in global temperature of different scenarios dealing with extreme poverty. Move extreme poor <1.90$ PPP per day above extreme poverty level. The green line shows the warming determined by ending extreme poverty over the baseline (yellow line). 50% (dark shading) and 66% (light shading) confidence ranges are obtained in MAGICC by running the model 600 times each time using a slightly altered set of climatic parameters that are based on historical observations. See "Methods" for more details

2010, which is roughly in line with similar estimates[26]. The global elite or top 10% of income earners (with incomes higher than 23$ PPP daily) are responsible for about 36% of global carbon emissions for their consumption of goods and services emitted in the production process along global supply chains. This would comprise mainly the populations of the rich countries such as the US, the EU, Japan, Australia, and Canada plus the elites of developing countries and transition economies. In comparison, the global poor or bottom 50% of income earners with income of <$2.97 PPP cause about 15% of global carbon emissions. The bottom half includes also the extreme poor of more than a billion people in 2010 earning <1.90$ PPP and contributing <4% of global carbon emissions.

This unequal distribution is also reflected in the respective carbon footprints for each global income category. The carbon footprint of the lowest income category is about 1.9 t $CO_2$-e whereas the global elite's carbon footprint amounts to 26.3 t $CO_2$-e per capita on average. The low direct carbon footprint of the lower income categories might be partly due to the omission of non-commercial biomass (see "Limitations" section for further discussion). Whereas the lower income households spend a larger share of their income on necessities such as food, clothing and shelter, with increasing income the share of luxury items, services and travel increases (Supplementary Fig. 1).

**Carbon implication of poverty alleviation.** Extreme poverty, as defined by the World Bank, has been cut in half since the 1990s. But there are still one in five people in developing countries living in extreme poverty. Extreme poverty is mainly found in fragile and conflict-affected countries, predominantly in Southern Asia and sub-Saharan Africa. While the SDGs set a goal for eradicating extreme poverty, they do not specify how this might be achieved. Thus to investigate the carbon implications of the first SDG for moving the population in extreme poverty to the next higher income level we do just that without assuming changes in income in the other household categories or changing technology. We take the consumption patterns and the associated carbon footprint of people in extreme poverty (i.e, <$1.90 PPP) and move them to the next income level and their associated consumption patterns for the group $1.90–2.97 PPP by 2030, with population growth at the growth rate of low income countries according to the United Nations projection[3]. In addition to increases in household consumption, we increase proportionally government expenditure and investment to allow for a necessary expansion of production capacity required to supply the increase in household demand.

Since long-term climate impacts depend on cumulative carbon emissions we then take the resulting additional carbon emissions over time and add them to a baseline scenario. As baseline, we use one of the greenhouse gases (GHG) emission trajectories, known as representative concentration pathways (RCPs), adopted by the IPCC for its fifth Assessment Report (AR5) in 2014 that is consistent with limiting temperature changes to below 2 °C above pre-industrial level, i.e., RCP2.6. The RCP2.6 emission scenario assumes that global annual GHG emissions peak sometimes before 2020 and decline substantially afterwards. We use RCPs

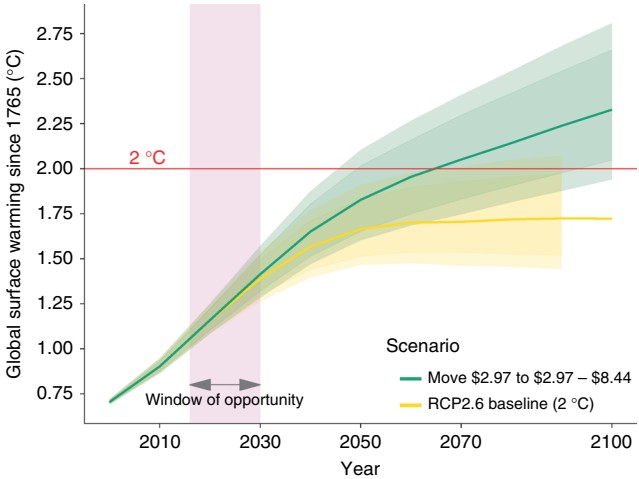

**Fig. 3** Predicted increases in global temperature of different scenarios moving poorest to the global middle class. Move global poor (<2.97$PPP per day) to next higher income category 2.97–8.44$ PPP per day. The green line shows the warming determined by moving the less-than-2.97 $/day-income category (i.e., the lowest half of the global population) to the group 2.97–8.44$ PPP/day over the baseline (yellow line). 50% (dark shading) and 66% (light shading) confidence ranges are obtained in MAGICC by running the model 600 times each time using a slightly altered set of climatic parameters that are based on historical observations

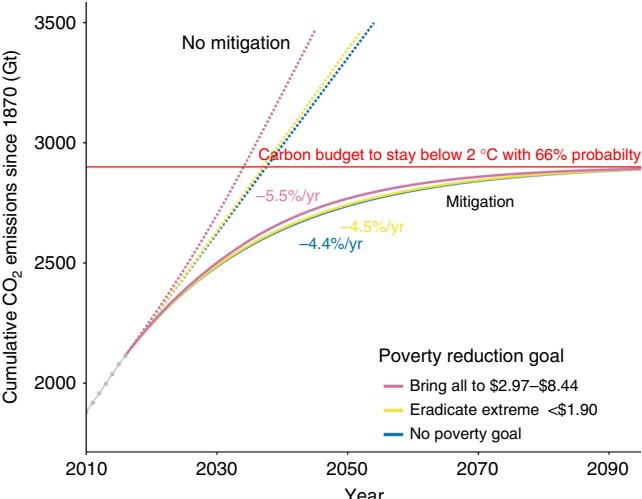

**Fig. 4** Historic cumulative global $CO_2$ emissions from 1870 through 2015 and predicted paths. Gray dots before 2015 shows historical cumulative $CO_2$ emissions since 1870 in Gigatons (Gt). Solid lines after 2015 shows predicted paths for each scenario and required annual reduction in carbon intensity in percent to stay below 2 °C above pre-industrial levels with 66% chance. Dotted lines after 2015 show predicted carbon emission paths if no mitigation measures are taken

because they are produced to be independent from underlying socio-economic assumptions as these projections can result from very different Integrated Models. To convert emissions scenarios into global-mean temperature we used MAGICC version 6. Model for the Assessment of Greenhouse Gas Induced Climate Change (MAGICC) is used in many scientific publications and integrated models. It is also used by the IPCC to facilitate integrated model comparisons. The results are presented in Figs. 2 and 3.

The good news is that lifting people out of extreme poverty has only relative little carbon implications with a projected increase of about 0.05 °C above the IPCC base run by the end of the 21st century. This relatively small increase makes intuitive sense given the low per capita carbon footprint of the extreme poor. However, the situation changes for a policy goal of not only eliminating extreme poverty, but also where we move people into, what may be considered as the global middle class (between the 50th and 75th income percentile), which is by the standards of industrialized countries a fairly modest income of between 2.97$ PPP and 8.44$ PPP per capita a day. For comparison, the middle class in the developed world is quite far removed from what we are referring to as the global middle class—in fact the global middle class is below the developed world's poverty line. When moving everybody to a higher income level we add another 0.6 °C by the end of the century. This is a fairly significant increase on top of the IPCC scenario given that total increase in global temperature since the industrial revolution is about 1 °C[27] that can significantly increase the required speed and extent of future emission reductions.

## Discussion

For a greater than 66% chance to keep average global temperature below 2 °C above pre-industrial levels, society can emit about 2900 Gt of $CO_2$ from 1870 (Climate Change 2014 Synthesis Report, p. 64, T 2.2[28]) or about 800 Gt $CO_2$ from 2017[29]. Figure 4 shows the historic cumulative global $CO_2$ emissions from 1870 through 2015 together with the predicted paths resulting from the combination of our scenarios with and without poverty reduction

goals (solid lines), with and without mitigation to stay below 2 °C above pre-industrial levels with 66% chance (dotted lines). Average annual $CO_2$ emissions reduction rates for the period 2017–2100 corresponding to each poverty reduction goal required to stay below 2 °C with 66% probability are also shown in Fig. 4. Using a standard simplified approach for the mitigation scenarios as, for example, in Jackson et al.[30], we find that the mitigation rates needed to stay within 2° are −4.4%/yr if we assume constant 2016 emission rates (36.4 Gt/year of $CO_2$, just 0.2% higher than 2015, Le Quéré et al.[29]) without poverty targets, −4.5%/yr if we add to the base the carbon needed to eliminate poverty (<$1.90/week), and −5.5%/year if we add to the baseline the carbon needed to bring people to the $2.97–8.44 range of expenditure per week. Specifically, the average annual mitigation rate for the incremental carbon emissions from eliminating extreme poverty is 2.8% higher (0.1 percentage point increase) than without poverty reduction goals. The mitigation rate for the additional carbon from bringing people to $2.97–8.44 per week range of income is 27.03% higher (1.1 percentage point increase) than without poverty reduction goals (Fig. 4).

However, these technological solutions could be hard to implement[31]. To meet the 2° target, even without the additional carbon needed to lift people countries out of poverty, the world already needs to decarbonize at an annual rate of more than 4% from now on depending on population and economic growth assumptions[32]. Only a handful of countries have been close to 4% historically. Sweden has had the biggest success by decarbonizing its economy at about 4% a year between 1970 and 1990 (IEA data), mostly by replacing fossil fuels-based power plants with nuclear and hydropower. France has managed to decarbonize at 3.8% with an analogous approach during the same period. More recent (2000–2014) global rates have been much lower (about 1.3%) and for most western economies (<2%) these numbers are inflated by a decreasing manufacturing sector and concomitant increased imports[33]. The International Energy Outlook 2016 predicts small reductions in energy intensity and carbon intensity over the 2012–2040 period. EIA reports that these forecasts are highly uncertain as dependent on policies and regulations that will be implemented, as well as the potential role of new

technologies. EIA data[18] show a declining ratio of $CO_2$ emissions to real GDP until about 2000, slowing down afterwards globally and remaining flat for non-OECD and non-Annex I parties. Since 2000 emissions per unit of energy use have, in fact, been rising, because of the greater use of coal. Most IPCC scenarios that leave decarbonization to later dates, require negative emissions relying on carbon capture and storage (CCS) technology that currently show little prospect of being commercially deployable[31]. There are numerous studies showing that technological advances have not been as successful as initially thought. As an example, most IPCC scenarios that have a good chance to keep the temperature below 2° require negative emissions, some requiring substantial implementation of this technology in the form of bio-energy production with CCS (i.e. bio-energy with carbon capture and storage or BECCS), as early as 2030 (IIASA scenario database). However, implementing energy production combined with CCS is proving much harder than expected[34]. According to the Global CCS Institute[35] there are 15 operational CCS projects around the world, capable of capturing up to 30 million tons per year of $CO_2$. Despite success stories in some countries to reduce emission intensities global carbon emissions have been on the increase (2% in the 1970–2014 period and 2.45% since 2000) and reductions in energy intensities or carbon intensity of fuel mix have been more than compensated by additional emissions in economic growth and lifestyle changes[36]. So far technology has not been able to keep up with additional emissions and our scenarios would require even more technological progress on top of what we would have otherwise (see "Methods" references therein). This increases the pressure on demand-side measures jointly with lifestyle and behavioral changes, to reduce the amount of energy required for transport, housing, etc. Given that the global elites with carbon footprints of more than 26.3 t per capita are responsible for 36% of the current carbon emissions a discussion on global income distribution and carbon intensive lifestyles should become part of the discourse of future efforts towards a low carbon society.

## Methods

**Relationship to earlier studies.** We use the consumption-based approach to carbon accounting. This is a different approach from earlier papers, say for example, by Chakravarty et al.[71] and Chakravarty and Tavoni[13] as our analysis is based on supply chain carbon emissions that go beyond household direct energy consumption, i.e., accounting for the carbon emitted during the production of consumption items and services, which are then allocated to the final consumer, and are based on very detailed information on income distribution and consumption patterns. This latter point also distinguishes our paper from Chancel and Piketty[26] who use GTAP multi-regional IO (MRIO) for data on countries incomes and emissions. But then, in order to move from country average emissions to emissions of different individual (income) groups within countries they allocate $CO_2$ to different income quantiles using country-year average decile income/consumption data from the World Bank. This is a top-down approach and does not use the detailed consumer expenditure surveys at the household level for different income groups as proposed here. The only paper we are aware of that addresses within country inequality at the global level (Dennig et al.[37]) uses a modified version of RICE one of the standard integrated assessment models (IAM), which is based on 12 regions and World Bank data on national income, not detailed consumption surveys. Most IAMs are based on very large regional aggregation and do usually not account for income inequality between countries. For example, MESSAGE–MACRO focuses on detailed complex "bottom-up" energy supply sectors but still uses 11 world regions and does not have equivalent micro detailed consumption at different income levels and developing countries and is not set up to address poverty or inequality issues. The same holds for iPETS which has 31 regions, however very aggregated when it comes to poorer countries (for example sub-Saharan Africa is one region). In contrast, we use 90 developing countries provided by the World Bank's Global Consumption Database and representative developed countries, including the US, 27 EU countries, Australia, and Japan (see below for more detail). That is not to discredit in any way the function and importance of the IAMs but they are built for other purposes and each of these approaches has its own strengths and weaknesses and each provides a different perspective on the policy options available to meet climate change targets. Extreme poverty is typically defined in terms of satisfying immediate basic needs such as food and shelter, needed for long-term physical well-being. Thus, we believe that

our approach using detailed household expenditure data, with great detail for a large number of developing countries, contributes a different and useful perspective.

**Carbon footprints and consumer expenditure surveys.** We use multi-regional input-output (MRIO) analysis to compute household carbon footprints for different income groups for 189 countries. The carbon footprint includes direct emissions associated with a household's activities such as heating, driving and using electricity as well as indirect emissions associated with the production of goods and services a household consumes[24, 25] as well as assumed associated shares of required investments and government expenditure to support increase in production capacity. The consumption patterns of different household groups in 90 developing countries are provided by the World Bank's Global Consumption Database[21] and households in developed countries are based on national consumer expenditure surveys reported by the respective national statistical offices. The global MRIO table is collected from the Eora database[38]. Eora is a multi-region IO database that provides a time series of high-resolution IO tables with matching environmental and social satellite accounts for 186 countries. This study focuses on 2010 to be consistent with the World Bank's Global Consumption Database.

In the World Bank's Global Consumption Database, households in developing countries are categorized in four consumption segments: lowest, low, middle, and higher consumption groups. They are based on global income distribution data, which rank the global population by income per capita, which we use and scale up by actual global population shares for each income category. The lowest consumption segment (below \$2.97 per capita a day) corresponds to the bottom half of the global population; the low consumption segment (between \$2.97 and \$8.44 per capita a day) to the 51th–75th percentiles; the middle consumption segment (between \$8.44 and \$23.03 per capita a day) to the 76th–90th percentiles; and the higher consumption segment (above \$23.03 per capita a day) to the 91st percentile and above (Supplementary Fig. 1). The monetary values are in PPP[39]. Purchasing power parities tells us how many dollars are needed to buy a dollar's worth of goods in a country as compared to the United States. In other words, PPP estimates the amount of adjustment needed of the exchange rate between countries in order for the exchange to be equivalent to each currency's purchasing power.

The first of the UN SDGs, which we are exploring in this study, is to move people out of extreme poverty by 2030, defined as <\$1.90 PPP per day. However, the World Bank global consumption database (WBGCD) provides only the consumption patterns for the group with consumption of <\$2.97 PPP per day. To estimate the carbon emissions per capita associated with consumption for the people spending <\$1.90 PPP per day, we first select the poorest countries with an average expenditure of <\$1.90 PPP per day or \$456 PPP per year and use their average per capita consumption data as representative of the poorest income category (<\$1.90 PPP per day) at the global level.

While the WBGCD represents the consumption patterns of the low income categories it is less representative for consumption patterns of higher income categories which represents consumers from developed countries. Thus in addition to the consumer expenditure surveys for 90 developing countries included in the World Bank's global consumption database (http://datatopics.worldbank.org/consumption/) we included consumer expenditure surveys from the US[22], the EU[23], Australia[24], and Japan[25]. Population data for different consumer groups were collected from the World Bank Povcalnet[4]. According to the PovcalNet database, developed countries only have a share of about 1% of the global population in the < \$8.44 consumption groups. In terms of global middle income (\$8.44–23.03), developed countries' share in this group accounts for about 19%, while their share of the global high consumer group (>\$23.03) is 89%. Therefore, we use the World Bank's 90 developing countries' consumption data (accounting for 89% of total population in developing countries) to estimate per capita carbon footprint for the extreme poor (<\$1.9 PPP per day), \$1.9–2.97 PPP per day, and \$2.97–8.44 PPP per day. To calculate the footprint for the \$8.44–23.03 group, we split the countries into two groups. We use the consumption expenditure data of this consumer group of the 90 developing countries in the World Bank's consumption database (representing 72% of the global total in that category; developing countries account for 81%) and consumer expenditure surveys from the US, Japan, Australia and EU to represent consumption patterns in developed countries (representing 16% of the global total in that category; developed countries account for 19% in this category); their combined share is 87% of the global total in that category. For the highest consumer group (>\$23.03) we used the average expenditure for people falling in that consumer category from the 90 developing countries (representing 8% of the global total in that category), the EU, Japan, Australia and the US, which represent about 73% of the global population in that category.

**The multi-regional input–output approach.** In a MRIO framework, different regions are connected through inter-regional trade. The technical coefficient sub-matrix $\mathbf{A}^{rs}$ consists of $\{a_{ij}^{rs}\}$ is given by $a_{ij}^{rs} = t_{ij}^{rs}/x_j^s$, in which $t_{ij}^{rs}$ represents the inter-sector monetary flows from sector $i$ in region $r$ to sector $j$ in region $s$; $x_j^s$ is the total output of sector $j$ in region $s$. The final demand matrix is $\{y_i^{rs}\}$, where $y_i^{rs}$ is the final demand of region $s$ for goods of sector $i$ from region $r$. Using matrix

notation and dropping the subscripts, we have

$$A = \begin{bmatrix} A^{11} & A^{12} & \cdots & A^{1n} \\ A^{21} & A^{22} & \cdots & A^{2n} \\ \vdots & \vdots & \ddots & \vdots \\ A^{n1} & A^{n2} & \cdots & A^{nn} \end{bmatrix}; Y = \begin{bmatrix} \sum y^{1r} \\ \sum y^{2r} \\ \vdots \\ \sum y^{nr} \end{bmatrix}; x = \begin{bmatrix} x^1 \\ x^2 \\ \vdots \\ x^n \end{bmatrix}$$

Consequently, the MRIO framework can be written as:

$$x = (I - A)^{-1} y \qquad (1)$$

where $x$ is a colum vector of sectoral total output for all countries; $(I\text{-}A)^{-1}$ is the Leontief inverse matrix which captures both direct and indirect economic inputs to satisfy one unit of final demand in monetary value; $I$ is the identity matrix with ones on the main diagonal and zeros everywhere else; $y$ is a column vector of sectoral total final demand for all countries.

To calculate the consumption-based $CO_2$-e emissions, we extend the MRIO table with a vector of sectoral $CO_2$-e emission coefficients for all regions, $k$:

$$k = \begin{bmatrix} k_1 & k_2 & \cdots & k_n \end{bmatrix}$$

Thus, the total consumption-based $CO_2$-e emissions for all regions can be calculated by:

$$CO_2\text{-e} = k(I - A)^{-1}Y + HH_C \qquad (2)$$

where $CO_2$-e is a row vector of the total $CO_2$-e emissions (both supply chain emissions and household direct emissions) associated with total final consumption ($Y$), sum of household consumption, government expenditure and stock change and investment, in all countries; $k$ is a row vector of $CO_2$-e emissions per unit of economic output for all economic sectors in all regions. $k (I\text{-}A)^{-1}Y$, a row vector, captures supply chain emissions of final demand of goods and services in different countries; $HH_C$ is a row vector of household direct emissions of all regions, e.g., driving and house heating;

Household direct emissions for each consumption group in different countries are estimated based on their household fuel consumption from the consumer expenditure survey data.

**Matching consumption survey to MRIO household consumption.** In the WBGCD[21], and the global multi-regional IO database[38] have been developed to enable global analyses and are uniform across countries. To match the consumption categories provided by the Global Consumption Database with Eora economic sectors we follow a well-established approach (e.g., refs. [40, 41]). We first assign consumption items from the WBGCD to different IO sectors and then scale the consumption survey data for each sector by four income groups to match the total household consumption of each sector from the global MRIO table (see Bridge Matrix in Supplementary Data). This method assumes that national consumption data is more accurate than the consumption survey data in terms of total consumption of each good or service. However, the survey date provides useful information about the distribution of goods and services to different income groups.

In the WBGCD database, all consumption items are categorized according to the International Comparison Program (ICP) classification, equivalent to the international Classification of Individual Consumption According to Purpose (COICOP). To estimate the carbon footprints of different consumption groups, we first map the ICP classification of consumption items to aggregate Eora IO sectors. Three situations can occur:

- One ICP category in the WBGCD corresponds to one Eora sector. In some cases there is a perfect match between an ICP category and an Eora sector. The mapping is straightforward.

- Multiple ICP categories correspond to one Eora sector. For example, in the ICP category, "Processed fish and seafood", "Cheese, butter and margarine", "Other edible oil and fats" can only fit to Eora sector Food & Beverages. Given the more detailed ICP classifications than the Eora sector, this allocation is the most frequent case in the matching process. Mapping is also straightforward in this case.

- One ICP category in the WBGCD corresponds to more than one Eora sector. There are some cases that one consumption item category may be produced from multiple economic sectors. For example, some food consumption categories can either be sold directly from the farm and thus would be linked to Eora sector "Agriculture" or have been processed and thus need to be linked to Eora sector "Food & Beverages". However, there is no good reference to split the aggregate consumption categories, such as "Fresh or Chilled Vegetables Other than Potatoes", into "Agriculture" and "Food &Beverages' sectors". And, the allocation of an aggregate consumption category might vary from country to country. Most studies ignore this problem[42] and just equally distribute the aggregate category in consumption (as, e.g., done by the World Bank's Global Consumption Database) or when linking environmental accounts to IO accounts (e.g., ref. [43]), or numerous IO studies linking consumer expenditure survey to IO categories (e.g., refs. [41, 44]). We followed this practice for this study but also added an uncertainty analysis using an allocation approach based on different possible "extreme" bridging matrices. We select the possible IO sectors that can be linked to the ICP sectors. For the

maximum value, we assign the ICP category to the sector with the highest emissions multiplier, for the lowest possible value, we assign the ICP category to the sector with the lowest emission multiplier. We then take these extreme or maximum possible deviations from the allocation we had chosen and calculate the deviation for each household group and show the range through error bars for each household category (Supplementary Fig. 2). Our uncertainty analysis shows that re-allocation of the consumption categories that may fall into multiple economic sectors has a relatively small impact on per capita footprints of different household groups. The uncertainty range of the per capita CF for all household groups is <2% between the max and the min and even less of an issue for our scenarios only involving the lowest two income categories.

**Other final demand sectors by consumer groups.** In addition to household consumption we also include other two final demand sectors, government expenditure and capital formation, to accommodate the necessary expansion of production capacity required to produce the increase in household demand. We allocate capital formation and government expenditure column vectors from EORA MRIO database to the four consumer groups based on the share of household consumption by different consumer groups in the total household consumption. Equation (3) was used for disaggregation of capital formation and government expenditure.

$$Cap_{Disagg} = diag(Cap) * HH_{share} \qquad (3)$$

where $Cap_{Disagg}$ is disaggregated capital formation by four consumer groups; $Cap$ is a vector of capital formation by sectors; and $diag(Cap)$ denotes a diagonal matrix with the elements of vector CAP on the main diagonal. $HH_{share}$ is a matrix showing the shares of consumption by different consumer groups in different sectors.

This method assumes that the higher the consumption for a certain product the higher the required capital formation in that sector. Lacking better information, we make the same assumption for government expenditure. For example, we allocate the government expenditure column to different consumer groups based on their consumption of other services from the consumer expenditure survey data.

**Scenario analysis.** In this study, we set up two scenarios for future additional carbon emissions due to moving poor people out of poverty by 2030 but these emissions are then continued until 2100 to be consistent with IPCC climate change projections. Scenario 1: lifting all people spending <$1.90 PPP per day (i.e., 840 million people[45]) to a higher consumption group spending between $1.90 and $2.97 PPP per day by 2030, equally spread over the years, with an average of 56 million people moving out of poverty per year. Scenario 2: lifting all people with a per capita income of <$2.97 PPP per day to a higher expenditure group spending between $2.97 per day and $8.44 per day by 2030 with an equal share of people annually moving to the next higher consumption group. We also incorporate population growth into the model. As population growth data for different consumption groups in different countries is not available, we use the UN population growth projection[3] for this study. Then, we use population growth rates by different country classifications (low income, lower middle income, upper middle income, and high income) to represent different growth rates in different consumption groups. For instance, we use the average growth rate in low income countries (i.e., <$1045 per capita per year) for the <$1.90 and $1.90–2.97 PPP consumption groups, while we use the average growth rate in lower middle countries (i.e., $1046–4125 per capita per year) to represent the growth rate for the $2.97–8.44 consumption group. In terms of changing fertility rates associated with higher incomes we assume that people moving up to a higher consumption group will have the same fertility rates as their new income peers, otherwise we strictly stick with the UN predictions.

Since long-term climate impacts depend on cumulative carbon emission we add the additional emissions needed to achieve the SDGs of moving people out of extreme poverty to a baseline emission pathway. (By additional emissions, we mean the higher emissions based on the new expenditure group minus previous emissions in the lower expenditure group.) As baseline, we use one of the GHG emission trajectories, known as Representative Concentration Pathways (RCPs), adopted by the IPCC for its fifth Assessment Report (AR5) in 2014[46]. RCPs are concerned with GHG concentration trajectories but the database provides, as a supplement, emission trajectories calculated as inverse emissions, i.e., implied emissions corresponding to the scenario concentrations, using the MAGICC model (see below). These standardized scenarios emphasize not only the long-term emission stabilization level but also the trajectory that leads to that outcome. We use the scenario that is consistent with limiting temperature changes to below 2 °C above pre-industrial level, i.e., RCP2.6. The RCP2.6 (now referred to as RCP3PD in the database, where "PD" stands for Peak and Decline and 3 refer to radiative forcing, in watts per square meter units, predicted by the year 2100) emission scenario assumes that global annual GHG emissions peak sometimes before 2020 and decline substantially afterwards. Other scenarios include RCP4.5, RCP6, and RCP8.5. These are all plausible pathways. We chose RCP2.6 here because the

research question we are addressing is whether meeting the SDGs can be achieved without affecting the climate goal[47].

RCP scenario data contain reference GHG emissions. However, the database does not report socio-economic variables underlying each scenario. We present un-abated emissions. To convert emissions scenarios into global-mean temperatures we used MAGICC version 6[48]. MAGICC is essentially a reduced-complexity model used in many scientific publications and integrated models. It is also often used by the IPCC, for instance, to facilitate integrated model comparisons. MAGICC integrates a set of simple models linking emissions to concentrations to global radiative forcing and to global changes in temperature and sea-level. For more details see refs. [49, 50]. 50 and 66% confidence ranges are obtained in MAGICC by running the model 600 times using a slightly altered set of climatic parameters that are based on historical observations. Details of this approach are available in[51].

**Limitations.** The United Nations[52] in a resolution adopted by the Assemble set 17 goals focusing on poverty alleviation, hunger, inequality and inclusion, gender, education, access to water and sanitation. Many of these so-called UN SDGs are related to income but the selected examples already show that poverty goes beyond simply income. This multi-dimensionality of poverty is well reflected in the literature recognizing that income measure alone are a poor representation of poverty[9, 10]. Albeit true, many of these ailments are linked to income especially at the lowest level of consumption where people spend most of their money on basic consumption items such as food, clothing and shelter (see also Supplementary Fig. 1 based on consumer expenditure patterns of the World Bank (WP)'s poverty database). In this study we focus explicitly on the first SDG and look at the implications of moving people to a higher expenditure group.

The WB database is based on consumer expenditure surveys for 90 developing countries and four expenditure categories and here supplemented by consumer expenditure surveys from developed countries. In our scenarios we move people from one expenditure category to the next (crossing the poverty line) assuming the consumption patterns of the higher expenditure group. The carbon emissions are based on greenhouse gas emissions as represented in the global MRIO database captured through market exchange. We account for carbon emissions associated with burning fuels such as char coal, crop residues and other biomass for cooking and heating when captured through market transactions in the consumer expenditure surveys. These fuel sources can be a substantial part of the livelihoods (e.g., ref. [53]). As incomes increase, biomass does not simple get replaced by other fuels but households tend to use a wide mix of fuels in their fuel transitions[54], we capture those transitions only when reflected in the consumer surveys. If rural poor replace biomass with paid-for energy than we ignore the potential positive impact on reduced deforestation. Similarly, we do not deal with other land use related carbon emissions associated with subsistence agriculture. At these low levels of income any additional income would not replace these forms of livelihood income but rather would help diversify their livelihoods[55].

In this study we do not account for land use related emissions. The literature on how poverty alleviation would change land use at a global level is quite inconclusive and really beyond what we could hope to achieve in this paper. Most of the studies that link land use to consumption patterns are at the regional scale[56] or at best at the national level[57], and frequently do not explicitly account for differences in income. The few global level studies show that land consumption seem to increase with higher income[58].

A somewhat related problem is the different impacts of meat consumption versus other food intake, which have quite different environmental impacts in terms of land use, but are more relevant for our study in terms of methane emissions. While we do account for these differences in terms of emissions, we do not do this in the best possible way as meat consumption is subsumed as part of consumption of agricultural products. Higher income people would pay higher prices for agricultural products and thus cause higher carbon emissions. In this sense we have captured differences in meat versus vegetable consumption. A better way would have been to have a more disaggregated global model to better capture different types of food consumption. Part of the problem is the trade-off of sectoral detail and country coverage of different global multi-regional IO models. For a recent comparison of various global MRIOs see ref. [59].

In this study, we make the assumption that the consumption bundles of each expenditure group remains the same until 2030, in particular for the low expenditure group included in our scenario analysis. While there is a huge amount of literature focusing on consumption patterns within income/expenditure categories this is less so at the global level as discussed in this paper; but the goal of the paper is not to predict carbon emissions for the year 2030 or 2100 but to investigate the implications of two poverty alleviation scenarios (of moving people from a lower category to a higher one) by 2030 using available detailed data on consumption for different expenditure and income categories from the World Bank and other statistical agencies. We want to investigate these counter factual scenarios based on detailed available data and do not want to "dilute" these scenarios by modifying existing consumption bundles in all income categories other than moving people from one category and the associated consumption bundle to the next. Supplementary Table 1 shows that there is considerable stability in terms of expenditure patterns across the poor in that three quarters of the expenditure is for food, shelter and housing, and the remaining expenditure categories only make small contributions to the expenditure.

This is also reflected in the literature. The most important determinant of the carbon footprint is income (see e.g., Minx et al.; Ahmad et al.[40, 60]), which we consider explicitly in our two poverty alleviation scenarios but other determinants of per capita carbon footprints such as urban characteristics[61], population density[62], lifestyles (Baiocchi et al.[41]), and household size[63], etc. are not considered in this study as these do not exist at the global level and most of the countries we include in our study, and are only available for specific countries or selected cities. In summary, the "constant consumption assumption" for the poor for the next decade or so is reasonable given that income and associated composition are the most important factors and we change them explicitly based on detailed available information; and that we are only interested in the counter factual poverty alleviation scenario and their carbon implications. Moreover, there is a lack of available studies for most countries considered in this study; and finally, we find a relative stability of consumption patterns of the poorest in poor countries.

Similarly, we use current emissions intensities (from Eora) until 2030, ignoring well-established hypothetical trends of reducing energy intensity and climate policy. This assumption would probably overestimate the poor's future emissions. Rather than explicitly modeling various technological assumption as is done by Rao[64] we calculate the additional carbon emissions of poverty alleviation and the required reduction in carbon emissions. These additional carbon emissions of poverty alleviation are added to the RCP2.6. The RCP are abstracted from possible socio-economic scenarios that can produce them in agreement with (e.g., Moss et al.[65]). According to the IPCC AR5: "The RCPs ARE NOT [emphasize added] associated with unique socioeconomic assumptions or emissions scenarios but can result from different combinations of economic, technological, demographic, policy, and institutional futures" (Wayne[66], p. 8). "RCPs each describe an emission trajectory and concentration by the year 2100, and consequent forcing. Each trajectory represents a specific synthesis drawn from the published literature. From this "baseline", researchers can then test various permutations of social, technical and economic circumstances." (Wayne[66], p. 9) And this is exactly how we are using the RCP as a reference emission trajectory to assess the carbon consequences of poverty alleviation. We do not make any assumption of whether innovations and technical change will deliver or not. We "just" assume, based on the IEA report, that in the next 12 years that might not be major changes in carbon intensity. We only calculate the additional gains in reduction required to offset the additional carbon emissions from poverty alleviation.

Another important component of the poverty alleviation scenario is the estimation of population growth. Estimation of population growth is ideally based on the population projections for each income group in each country until 2100. However, these data are not available. On the other hand, we understand that different countries may have different population growth rates and using the average growth rate for low income countries for the low consumption group may lead to relatively large uncertainty. Given the predictions available from the UN World Population Prospects[3] we have two choices: (1) to use predictions that represent income segments and thus requires averaging across countries (the average population growth rate for countries within an income bracket) based on estimates by the UN, for example, we use the average population growth rate of low income countries to represent the population growth for the extreme poverty group and the consumption group of >$2.97 per day, and use the population growth rate of lower middle income countries for the consumption group of $2.97–8.44 per day; or (2) to use country specific predictions but ignore the changing composition of expenditure groups (population growth rate by country). Both estimates have shortcomings. Applying the national average growth rate to the low income group within a country may lead to an underestimation of population growth for low income groups as their growth rate might be higher than the national average[67]. However, using the average population growth rate across all low income countries may also lead to uncertainty because countries have different population growth rates even though they all fall into the low income country category. In this study, we use option (1) i.e., using the average growth rate of the respective country group mainly because the country specific predictions are not an option to us anymore after we introduce our poverty alleviation scenarios as these would significantly influence income and thus fertility and mortality rates of these low income countries and the country specific predictions by the UN would not be applicable to these countries anymore. When comparing the two approaches, we find that option 2 provides 13% lower carbon emissions under scenario 1 and 18% lower under scenario 2 than by using option 1.

The scaled investment and government expenditure is likely to have some benefits for the poor but it is hard to estimate which benefits they accrue. For example, one might argue that a lot of the infrastructure investment benefits more the middle class, who can extract greater benefit from such investments. A simple example is road-building which leads to increased vehicular mobility. Those in extreme poverty have little to benefit from this road-building. We thus see these expanded government expenditure and investment as accompanying the expansion in producing those additional goods and services associated with poverty alleviation but they are not the trigger of poverty alleviation. These initial transfer payments are not explicitly included in this study.

There are a number of ways that government expenditure can lead to poverty alleviation such as direct payment transfer (cash transfer) which are being trialed or implemented in Brazil[68] and India[69], and direct and indirect subsidies (e.g., fuel subsidies, food subsidies, poverty programs)[72, 73]. However, we did not include any such payment transfer and/or subsidies or similar mechanism to increase the

income of the poor as outlined in these scenarios due to the complexity of these wealth distribution mechanism. However, this problem is partially mitigated by the use of expenditures instead of actual income, which is hard to measure in poorer countries. There is evidence that households in poorer countries have negative savings as they appear to spend more than they earn (see, e.g., Fesseau and van de Ven[70]). The fact that these types of transfer are implicit in the expenditure data is the standard justification for choosing expenditure over income in many studies.

**Code availability**. Programming code for MRIO analysis is available from the corresponding author on request.

**Data availability**. All data used in this study is from open access sources, including the World Bank Global Consumption Database, EORA MRIO database, the World Bank Povcalnet and the UN Population Prospects. The websites for data download are provided in the method section.

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

## Acknowledgements

This work benefited from support from the National Socio-Environmental Synthesis Center (SESYNC)—NSF award DBI-1052875. K.F. acknowledges the National Natural Science Foundation of China—NSFC 71628301. K.H. was partly supported by the Czech Science Foundation under the project VEENEX (GA ČR no. 16-17978S). We would like to thank Sandra Kadungure, Raúl Muñoz, Ana Ivelisse Sanchez, Laixiang Sun and Jinjun Xue for their suggestions and help with the manuscript.

## Author contributions

K.H., G.B., K.F. designed the research. K.H., G.B., K.F., performed analysis. All authors contributed to writing the paper.

## Additional information

**Competing interests:** The authors declare no competing financial interests.

