## [Peer Review File · Nature Communications]

Reviewers' comments:

Reviewer #1 (Remarks to the Author):

The article discusses the poverty-climate nexus, spotlighting the potential incoherence between climate targets and the Global Goals on poverty. Authors use EORA MRIO data World Bank consumption data in developing countries to assess the impact of poverty alleviation in developing world on global climate targets. They find that alleviating poverty under BAU is inconsistent with global climate targets.

The question raised in this paper is of great importance, and the method proposed novel enough to deserve attention, however I have sub concerns with the methodology used and the conceptual foundations of the analysis.

Major points

Ensuring consistency in the measurement of the global CO₂ and income/consumption distribution

The consumption dataset used in the paper (WBGCD) is based on consumer expenditure surveys for 90 developing countries. However, the paper presents estimates for a global distribution of CO₂e emissions (all individuals in the world) including estimates for the emissions of top 10% income earners (i.e. those emitting above 27.5tCO₂e per capita on average, according to the paper). My major problem here is that top 10% income earners include both developed and developing world individuals and bottom 90% earners also include developed world individuals (Lakner and Milanovic, 2015). More precisely, according to Lakner and Milanovic, more than 500 million individuals from mature economies belong to the bottom 90% global income earners, out of a total population of mature economies of 1.06 billion people. This implies that the estimates measured for the bottom 90% emitters is either missing 50% of mature economies' population or that it is not fully consistent (i.e. a methodology different to the one used for developing countries is used for developed countries emissions. This a major limitation to the methodology which is not discussed in the paper. In any case, it seems to me that it is not possible to derive a global distribution of CO₂e emissions from the World Bank dataset – focusing solely on developing countries.

Role of capital investments and government expenditure

The authors warn that their estimates do not account for government expenditure and capital investments. The key issue here is that eradicating poverty does require government expenditure and capital investments – some will favour capital investments over government spending, and other will support the opposite, but the important point is poverty eradication historically depends on one or the other of these two pillars.

Not taking such investments into account – and their associated CO₂e emissions- in the analysis is problematic from a conceptual point of view : it amounts to assuming that consumption patterns will evolve solely through market forces – but in a strange market, i.e. one without capital investments.

From a measurement point of view, taking into account such estimates is likely to modify the relatively optimistic numbers presented in Figure 2 (i.e. extreme poverty can be alleviated under BAU via an increase of only 0.1°C). In order to answer the important research question raised by the authors it is necessary to take the material impacts of these two dimensions into accounts. If these are not taken into account, an important part of the problem remains.

Description of the bridging procedure and quantitative data to assess it

The paper details the standard MRIO approach and briefly explains the procedure followed to bridge consumption survey data with MRIO data. The bridging process is crucial, as it is likely to but does not provide any quantitative assessment of this bridging process to show readers that we can actually be confident we the results presented. We would want some information in a Supplementary Material to be able to assess this.

Other points

Overall, the paper would benefit from further grammatical review to fluidify the style and correct some syntactical errors.

Line 72 : Citing Rockstrom J, et al. does not seem appropriate here.

Line 72 : Typo. « In not only »

Line 127 : Inconsistencies in notation between Figure 1 and this line (ton CO₂-e vs. CO₂eq)

Line 139 : Sentence requires syntactic revision.

Line 175 : Detail which technologies, we need to be convinced that this factor 10 reduction in the increase is possible.

Line 181 : Decarbonization rates are at odds with the method proposed in the paper. Sweden did decarbonize, as did France, its territorial emissions, but according to several MRIOs, it did not succeed to do so for its consumption-based emissions.

Line 228 : The Millenium Sustainability Goal does not exist to my knowledge.

Line 270 : Typo. « survey date »

Line 275 : The acronym WBGCD should come earlier. The sentence requires revision.

References : issue with « (ed[^](eds)). »

References

Lakner, C., Milanovic, B. (2015) Global Income Distribution: From the Fall of the Berlin Wall to the Great Recession, World Bank Economic Review

Reviewer #2 (Remarks to the Author):

Review of "Poverty eradication in a carbon constrained world"

The paper uses a global consumption survey to estimate the carbon emission of various income groups in the world and the impacts of reducing poverty on global emissions. They test two scenarios, one in which extreme poverty is eradicated (the goal of the World Bank and international community) and one in which all people are given the income of the global middle class. They find that eradicating extreme poverty does not threaten the objective of stabilizing climate change below 2C. They also find that bringing everybody to the level of the global middle class would threaten this objective, with current technologies. They discuss the implication for achieving the SDG1 and the objective of the Paris Agreement.

This is a very important paper, with innovative, relevant, and solid results. There is no doubt for me that the paper should be published.

In particular, it complements very usefully recent World Bank reports (Hallegatte et al. 2016; Fay et al. 2015) that make the case that climate mitigation policies can be introduced without slowing down poverty reduction, but do not calculate the impact of doing so on global emissions. And it adds to previous estimates that were not based on consumption surveys and were therefore unable to account for intra-country heterogeneity. I think the paper should have a lot of influence on the debate on climate change and development.

There is one thing that needs to be corrected before the paper is published, however. It relates to how your technological assumptions are presented, and to your conclusion regarding technological fixes.

** Technology. Technology assumptions play a major role in the paper, and this needs to be clarified in a few places:

- Line 55: "growth in energy use is strongly coupled to economic growth and poverty reduction", I think you mean correlated, and this is due to the technology that are used, and the consumption patterns (see (van Benthem 2015)). With different technologies and consumption patterns, this could change, so I would replace "coupled" by "correlated" and add that this relationship is not a law of physics that be remain valid over time: it can change, and it has changed in the past. See Box 1 in (Kalra et al. 2014) that shows how the energy intensity of GDP growth changed after the oil shock in 1973.

- Line 109-112: "This assumes that additional production [...] is met with current capacity": this is impossible and new capacities will have to be added. You want to say that this assumes that additional production is produced with similar technologies and similar energy intensity and similar carbon intensities. Line 112 you say that this leads to "conservative estimates" but the meaning is unclear: do you mean that you underestimate or overestimate emissions? I think you overestimate emissions because future technologies are likely to be more efficient than current one (see how a growing fraction of energy production comes from renewable in developing countries, and the empirical evidence in (van Benthem 2015)).

- A good complement to your approach would be to look at how technologies need to change to achieve different poverty reduction goals without threatening climate objectives (this is what is done in (Rozenberg et al. 2015) with economic growth).

Overall, I think the fact that current technologies do not allow to bring everybody to the level of the middle class without threatening the climate objective does not mean that it is impossible to do so – it only means that technologies need to improve (which we know already). So your first message (even with current technology, the eradication of extreme poverty is not an issue) is much more robust than the second one (it's more difficult to bring people to middle class level).

I think that the abstract and text is misleading when it suggests that bringing people to middle class level would be a problem for emissions – you have to clarify that this is with current technologies and lifestyles. What you show, therefore, is that policies are needed. (In contrast, you show that extreme poverty can be eradicated without big impact on emission, even in the absence of climate policies.)

And I think that the conclusion that your results do not show that a “focus beyond technical fixes” is needed. To say that you would have to replace the energy and carbon intensity used in the projections by estimates of optimistic future intensities to show that technologies do not have enough potential. You are not doing this analysis and therefore you cannot conclude on the fact that technology is not enough. (Note that I’m not saying that this conclusion is wrong – only that your analysis does not support it.) This needs to be corrected in the abstract and in the text.

** Poverty line. The paper makes multiple references to the \$1.25 poverty line, which is based on 2005 PPP exchange rate. Only a note in the figure mentions the change to the \$1.90 line with the 2011 PPP. It would be good to clarify the paper (which line is used and why). The two lines are designed to be equivalent so it is not a problem, but the exposition needs to be clarified. (And the abstract should not say that extreme poverty is currently represented by the \$1.25 line – it is represented by the \$1.90 line).

** Studies of distributional impacts of fossil fuel subsidies. You are right that few studies have been published on the distributional impacts of climate policies (even though you are missing a few, including (Rao 2013; Speck 1999; Metcalf 2008; Flues and Thomas 2015) and the discussion in (Fay et al. 2015). But a lot has been written on fossil fuel subsidies and their distributional impacts: similar arguments were made on these subsidies (removing them would increase poverty) but analysis of household surveys showed that removing them can reduce poverty is part of the savings are used for poverty reduction actions (see reviews in Fay et al. 2015; Ruggeri Laderchi 2014; IMF 2013; Arze del Granado, Coady, and Gillingham 2012).

** Population scenarios. While I’m comfortable with the assumptions that the authors need to do to cope with imperfect data, I think they should explain better the change in population. If I understand well, country-level population changes are applied for income categories within countries, and there is no economic growth beyond the change in income linked to bringing people out of poverty. For a scenario, it looks really strange. What should be done is to run a scenario with population change and economic growth, and to take all the poor people in this scenario and to make them non-poor. That’s complicated because there are only a few scenario for future poverty (the World Bank has the scenarios used in the Shock Waves report on the link between climate change and poverty, but they only go to 2030). One option for the authors is to explain that they focus on the incremental emissions from poverty reduction, compared with a scenario with no poverty reduction (basically, assuming that the RCP2.6 scenarios they look at does not reduce poverty at all – and compare with the same scenario with poverty reduction). I recommend they clarify this part.

** Costs. Line 188-190: “Other more proven technologies [...] not be affordable by poorer countries.” I think this statement is too strong. The world failed to provide electricity to Africa even with massive investments and using cheap fossil fuels, because of issues related to grid development and energy system management. The (small and decreasing) additional cost of renewable may well be more than compensated by how it makes it possible to provide decentralized energy, mitigating the issues related to system management and development of a national grid in low density countries. This statement should be changed.

** Effect on deforestation and other emissions. Line 376-378: I do not really understand this point, and why the authors need it. But eradicating poverty would change livelihood, and hopefully will reduce extraction from ecosystems and therefore deforestation, so that the argument does not really work. I think the authors have to acknowledge that they cannot capture these effects.

Deforestation hopefully would go in the right direction (less deforestation with higher income), so it should be okay. The authors also miss other aspects such as meat consumption, which could have large impacts.

** Other things

Line 228: "Millennium Sustainability Goal" is a nice mix of MDG and SDG!

Line 339: Sentence is not complete.

References

Arze del Granado, Francisco Javier, David Coady, and Robert Gillingham. 2012. "The Unequal Benefits of Fuel Subsidies: A Review of Evidence for Developing Countries." *World Development* 40 (11): 2234–48. doi:10.1016/j.worlddev.2012.05.005.

Fay, Marianne, Stephane Hallegatte, Adrien Vogt-Schilb, Julie Rozenberg, Ulf Narloch, and Tom Kerr. 2015. "Decarbonizing Development: Three Steps to a Zero-Carbon Future." Washington, DC: World Bank. <https://openknowledge.worldbank.org/handle/10986/21842>.

Flues, Florens, and Alastair Thomas. 2015. "The Distributional Effects of Energy Taxes." OECD Taxation Working Papers. Paris: Organisation for Economic Co-operation and Development. <http://www.oecd-ilibrary.org/content/workingpaper/5js1qwkqrbv-en>.

Hallegatte, Stephane, Mook Bangalore, Laura Bonzanigo, Marianne Fay, Tamaro Kane, Ulf Narloch, Julie Rozenberg, David Treguer, and Adrien Vogt-Schilb. 2016. "Shock Waves: Managing the Impacts of Climate Change on Poverty." *Climate Change and Development Series*. Washington, DC: World Bank.

IMF. 2013. *Energy Subsidy Reform – Lessons and Implications*. International Monetary Fund.

Kalra, Nidhi, Stephane Hallegatte, Robert Lempert, Casey Brown, Adrian Fozzard, Stuart Gill, and Ankur Shah. 2014. *Agreeing on Robust Decisions: New Processes for Decision Making under Deep Uncertainty*. Policy Research Working Papers. The World Bank. <http://elibrary.worldbank.org/doi/abs/10.1596/1813-9450-6906>.

Metcalf, Gilbert E. 2008. "Designing a Carbon Tax to Reduce US Greenhouse Gas Emissions." *Review of Environmental Economics and Policy*, ren015.

Rao, Narasimha D. 2013. "Distributional Impacts of Climate Change Mitigation in Indian Electricity: The Influence of Governance." *Energy Policy* 61 (October): 1344–56. doi:10.1016/j.enpol.2013.05.103.

Rozenberg, Julie, Steven J Davis, Ulf Narloch, and Stephane Hallegatte. 2015. "Climate Constraints on the Carbon Intensity of Economic Growth." *Environmental Research Letters* 10 (9): 95006. doi:10.1088/1748-9326/10/9/095006.

Ruggeri Laderchi, Caterina. 2014. "Transitional Policies to Assist the Poor While Phasing out Inefficient Fossil Fuel Subsidies That Encourage Wasteful Consumption." Contribution by the World Bank to G20 Finance Ministers and Central Bank Governors. <http://www.worldbank.org/content/dam/Worldbank/document/Climate/Transitional-Policies-Assist-Poor-Phasing-Out-Inefficient-Fossil-Fuel-Subsidies.pdf>.

Speck, Stefan. 1999. "Energy and Carbon Taxes and Their Distributional Implications." *Energy Policy* 27 (11): 659–67. doi:10.1016/S0301-4215(99)00059-2.

van Benthem, Arthur. 2015. "Energy Leapfrogging." *Journal of the Association of Environmental and Resource Economists* 2 (1): 93–132.

Reviewer #3 (Remarks to the Author):

I reviewed an earlier version of this article. The article has been revised substantially to include requested details on methodology and other suggestions. However, the analysis and its presentation remain unchanged, leaving many of the previously raised issues concerning troubling assumptions unaddressed. The conclusions at a qualitative level are relatively predictable, and their quantitative implications not reliable due to these assumptions. Nevertheless, given the importance of this question and that this is the first (known) attempt to provide an answer with a fair amount of work done, I would suggest the editor consider a revision subject to two important conditions: the authors include a new subsection on analysis caveats/assumption that brings together the problematic assumptions all together, their implication (direction of impact) for the conclusion for each assumption and if possible the overall estimates, and sensitivities where possible (as indicated below). Second, the translation of economic activity to climate sensitivity is seriously flawed (as described later) – the authors should stick to the metric of GHG emissions, drawing implications for aggregate impacts based on relative footprints of different income groups.

Assumptions in calculating GHG emissions:

- The authors in their baseline use current emissions intensities (from EORA) for the rest of the century, ignoring well-established trends of reducing energy intensity and climate policy. This assumption would probably overestimate the poor's future emissions.
- The authors assume the consumption pattern of the low-income segment would remain the same throughout the century – technology stays constant. This is also unlikely, though it is unclear in which direction the inaccuracy lies.
- The analysis ignores capital investment and government expenditure, though the authors are transparent about this. However, why not make an assumption about the footprint of both categories based on historical data, and propagate that into the future? This would be entirely consistent with the nature of the analysis, and provide a more complete assessment of footprint.
- Authors ignore land-use based emissions, about which they are also transparent. However, they can provide more qualitative insight on the quantitative implications going forward, if there is no basis to do sensitivities.
- The authors use the average consumption pattern and population growth of low-income countries as a proxy for those of low-income people, including those in higher income countries. This is likely to be an overestimate. Low income countries also have elites with high shares of total consumption – these countries' average expenditure patterns likely do not resemble those of the poor in wealthier countries.

Flaws in calculation of temperature impact:

- The authors add their emissions to an RCP2.6 trajectory of GHG concentrations, which make little sense. A socioeconomic pathway that yielded a 2.6-type emissions pathway would require substantial climate policy with transformative shifts in technology. This is entirely incompatible with a snapshot footprint of the poor from today. In an RCP2.6 world, the poor's emissions may be entirely negligible. If at all, authors should use a BAU type RCP trajectory, and carefully interpret the result as a sensitivity from an increase in emissions akin to the level of emissions emitted from poverty eradication.

Alternatively, and more accurately, the authors should estimate the aggregate emissions impacts (only) of relative differences in the per capita footprint composition for income groups, which is really the meat of the contribution here. In that vein, authors should in any case show direct and indirect footprints by income groups in some form.

Smaller points:

- the authors confuse wealth for consumption towards the end of the article
- the added paragraph on multidimensional poverty is welcome, but the statement about income being a good proxy for the poor is incorrect. Better to leave the discussion as an open caveat.

Reviewers' comments:

Reviewer #1 (Remarks to the Author):

The article discusses the poverty-climate nexus, spotlighting the potential incoherence between climate targets and the Global Goals on poverty. Authors use EORA MRIO data World Bank consumption data in developing countries to assess the impact of poverty alleviation in developing world on global climate targets. They find that alleviating poverty under BAU is inconsistent with global climate targets.

The question raised in this paper is of great importance, and the method proposed novel enough to deserve attention, however I have sub concerns with the methodology used and the conceptual foundations of the analysis.

Major points

Ensuring consistency in the measurement of the global CO₂ and income/consumption distribution

The consumption dataset used in the paper (WBGCD) is based on consumer expenditure surveys for 90 developing countries. However, the paper presents estimates for a global distribution of CO₂e emissions (all individuals in the world) including estimates for the emissions of top 10% income earners (i.e. those emitting above 27.5tCO₂e per capita on average, according to the paper). My major problem here is that top 10% income earners include both developed and developing world individuals and bottom 90% earners also include developed world individuals (Lakner and Milanovic, 2015). More precisely, according to Lakner and Milanovic, more than 500 million individuals from mature economies belong to the bottom 90% global income earners, out of a total population of mature economies of 1.06 billion people. This implies that the estimates measured for the bottom 90% emitters is either missing 50% of mature economies' population or that it is not fully consistent (i.e. a methodology different to the one used for developing countries is used for developed countries emissions. This a major limitation to the methodology which is not discussed in the paper. In any case, it seems to me that it is not possible to derive a global distribution of CO₂e emissions from the World Bank dataset – focusing solely on developing countries.

RE: yes, we agree with the referee that the World Bank global consumption database is not enough for the calculation of the global distribution of CO₂e emission (as used in figure 1). We did in fact use other data sources but did not make this sufficiently clear in the previous version, where we discussed the World Bank database but did not refer explicitly to the use of other data sources to represent lifestyles and consumption expenditure patterns for high income households. In fact, we used the expenditure survey data for different developed countries, including the EU,

Japan, Australia and the US, for the calculation of CO₂e emissions of the higher income groups. We added more text in the method section to explain how we derived carbon footprints for different income groups in developed countries to estimate the distribution of CO₂e emissions among household categories and countries. The added section reads as follows: “While the WBGCD represents the consumption patterns of the low income categories it is less representative for consumption patterns of higher income categories which represents consumers from developed countries. Thus in addition to the consumer expenditure surveys for 90 developing countries included in the World Bank’s global consumption database (<http://datatopics.worldbank.org/consumption/>) we included consumer expenditure surveys from US²⁹, the EU³⁰, Australia³¹ and Japan³². Population data for different consumer groups were collected from the World Bank Povcalnet⁴. According to the World Bank PovcalNet database, developed countries only have a share of about 1% of the global population in the less than \$8.44 consumption groups. In terms of global middle income (\$8.44- \$23.03), developed countries’ share in this group account for about 19%, while their share of the global high consumer group (> \$23.03) is 89% (see figure below). Therefore, we use the World Bank’s 90 developing countries’ consumption data (accounting for 89% of total population in developing countries) to estimate per capita carbon footprint for the extreme poor (<\$1.9 PPP per day), \$1.9 - \$2.97 PPP per day, and \$2.97 - \$8.44 PPP per day. To calculate the footprint for the \$8.44-\$23.03 group, we split the countries into two groups. We use the consumption expenditure data of this consumer group of the 90 developing countries in the World Bank’s consumption database (representing 72% of the global total in that category; developing countries account for 81%) and consumer expenditure surveys from the US, Japan, Australia and EU to represent consumption patterns in developed countries (representing 16% of the global total in that category; developed countries account for 19% in this category); their combined share is 87% of the global total in that category. For the highest consumer group (>\$ 23.03) we used the average expenditure for people falling in that consumer category from the 90 developing countries (representing 8% of the global total in that category), the EU, Japan, Australia and the US, which represent about 73% of the global population in that category. ”

Additional links and references:

BLS. Consumer Expenditure Survey. Bureau of Labor Statistics (2012).

Eurostat. Structure of consumption expenditure by income quintile. Statistical Office of the European Communities (2016).

Statistics Bureau Japan. National Income and Expenditure Survey for one-person households. Statistics Bureau Japan (2009).

Role of capital investments and government expenditure

The authors warn that their estimates do not account for government expenditure and capital investments. The key issue here is that eradicating poverty does require government expenditure and capital investments – some will favour capital investments over government spending, and other will support the opposite, but the important point is poverty eradication historically depends on one or the other of these two pillars.

Not taking such investments into account – and their associated CO₂e emissions- in the analysis is problematic from a conceptual point of view : it amounts to assuming that consumption patterns will evolve solely through market forces – but in a strange market, i.e. one without capital investments.

From a measurement point of view, taking into account such estimates is likely to modify the relatively optimistic numbers presented in Figure 2 (i.e. extreme poverty can be alleviated under BAU via an increase of only 0.1°C). In order to answer the important research question raised by the authors it is necessary to take the material impacts of these two dimensions into accounts. If these are not taken into account, an important part of the problem remains.

RE: We agree with the referee that emissions associated with government expenditure and capital investments is crucial for the estimation of additional emissions due to poverty

eradication. In the revised manuscript, we include all CO₂e emissions associated with household consumption, government expenditure and capital investments and provide more detail on the calculation of emissions associated with these final demand sectors in the method section. Our new results show that overall carbon footprint for different income groups increased compared with the extreme poverty alleviation scenario when only accounting for increases in household consumption, but this does not change our conclusion that lifting people out of extreme poverty may only cause a relatively small increase in temperature. But the picture changes significantly for the 'global middle income scenario' scenario (i.e. moving people to the above-\$ 2.97PPP expenditure category).

Description of the bridging procedure and quantitative data to assess it

The paper details the standard MRIO approach and briefly explains the procedure followed to bridge consumption survey data with MRIO data. The bridging process is crucial, as it is likely to but does not provide any quantitative assessment of this bridging process to show readers that we can actually be confident we the results presented. We would want some information in a Supplementary Material to be able to assess this.

RE: The methods section contains a description of matching WB's International Comparison Program (ICP) classification categories with EORA sectors. This is standard procedure used in numerous input-output studies. We have added a few references of seminal papers applying this approach (e.g. (Kok, Benders et al. 2006, Ornetzeder, Hertwich et al. 2008, Weber and Matthews 2008, Druckman and Jackson 2009, Baiocchi, Minx et al. 2010; Steen-Olsen, 2016). Following the referee's suggestion, we also included the bridging matrix in the supplementary data of this manuscript.

Additional References:

- Baiocchi, G., et al. (2010). "The Impact of social factors and consumer behavior on carbon dioxide emissions in the United Kingdom." *Journal of Industrial Ecology* 14(1): 50-72.
- Druckman, A. and T. Jackson (2009). "The carbon footprint of UK households 1990–2004: A socio-economically disaggregated, quasi-multi-regional input–output model." *Ecological Economics* 68(7): 2066-2077.
- Kok, R., R. M. J. Benders and H. C. Moll (2006). "Measuring the environmental load of household consumption using some methods based on input–output energy analysis: A comparison of methods and a discussion of results." *Energy Policy* 34(17): 2744-2761.
- Ornetzeder, M., et al. (2008). "The Environmental Effect of Car-free Housing: A Case in Vienna." *Ecological Economics* 65(3): 516-530.
- Steen-Olsen, K., Wood, R. and Hertwich, E. G. (2016), *The Carbon Footprint of Norwegian Household Consumption 1999–2012*. *Journal of Industrial Ecology*, 20: 582–592.
- Kok, R., et al. (2006). "Measuring the environmental load of household consumption using some methods based on input–output energy analysis: A comparison of methods and a discussion of results." *Energy Policy* 34(17): 2744-2761.
- Weber, C. L. and H. S. Matthews (2008). "Quantifying the global and distributional aspects of American household carbon footprint." *Ecological Economics* 66(2-3): 379-391.

Other points

Overall, the paper would benefit from further grammatical review to fluidify the style and correct some syntactical errors.

RE: done

Line 72 : Citing Rockstrom J, et al. does not seem appropriate here.

RE: thanks for spotting this. The appropriate text to this citation was closer to the end but we have removed the mentioning of the 'planetary boundaries' and associated reference.

Line 72 : Typo. « In not only »

RE: done

Line 127 : Inconsistencies in notation between Figure 1 and this line (ton CO₂-e vs. CO₂eq)

RE: fixed.

Line 139 : Sentence requires syntactic revision.

RE: the long sentence has been broken up and turned into 3 sentences.

Line 175 : Detail which technologies, we need to be convinced that this factor 10 reduction in the increase is possible.

RE: We have removed this statement as we do not calculate any abatement or technological change in this paper. The factor 10 reduction was based on technology assumption used in IPCC scenarios. Here we don't make any such assumptions. We add the additional carbon associated with moving people out of poverty until 2030; we then calculate the temperature increase associated with higher carbon emissions driven by increased consumption and associated increase in production and production capacity. These changes are introduced annually until 2030, in line with the SDG. We keep technology, i.e. carbon intensity constant until 2030. Beyond 2030, we do not make any assumptions about technical change but rather ask which annual reduction in carbon dioxide emissions is required to compensate for the additional carbon emitted through higher consumption levels. Using a simplified mitigation scenario analysis (see methods and references therein) we find that to abate these additional carbon emissions would require annual reduction in mitigation rates of more than 1 percentage point (an improvement of 27.03% over the baseline) stay below 2°C with 66% probability (see figure below and details in the methods section). We then ask whether this is feasible given current trajectories and technological progress and current technologies in the pipeline. We also believe that given the short span until 2030 this is an acceptable assumption justified also by

available forecasts (and their difficulty given policy uncertainty) and previous trends. For example, *The International Energy Outlook 2016* has forecast up to 2040 predicts small reductions in energy intensity and carbon intensity over the 2012-2040 period. EIA reports that these forecasts are highly uncertain as dependent on policies and regulations that will be implemented, as well as the potential role of new technologies. EIA data³⁷ shows a declining ratio of CO₂ emissions to real GDP until about 2000, slowing down afterwards globally and remaining flat for non-OECD and non-Annex I parties.

Details of the analysis are included in the methods section.

Additional reference:

EIA. International Energy Outlook 2016 - With Projections to 2040. U.S. Energy Information Administration (2016).

Line 181 : Decarbonization rates are at odds with the method proposed in the paper. Sweden did decarbonize, as did France, its territorial emissions, but according to several MRIOs, it did not succeed to do so for its consumption-based emissions.

RE: RE: We removed the decarbonization and associated statements. See also response for line 175 above. We still do point out that technical change, despite those listed examples, has not been able to keep track with increase in emissions. We also refer briefly to the outsourcing problem (i.e. consumption-based emissions being larger than production-based emissions: “... More recent (2000-2014) global rates have been much slower (about 1.3%) and for most western economies (less than 2%) inflated by a decreasing manufacturing sector and

concomitant increased imports.” EIA data shows a declining ratio of CO2 emissions to real GDP until about 2000, slowing down afterwards globally and remaining flat for non-OECD and non-Annex I parties.

Line 228 : The Millenium Sustainability Goal does not exist to my knowledge.

RE: thanks. We changed it.

Line 270 : Typo. « survey date »

RE: fixed.

Line 275 : The acronym WBGCD should come earlier. The sentence requires revision.

RE: we changed the sentence and introduce acronym earlier

References : issue with « (ed^(eds). »

RE: fixed.

References

Lakner, C., Milanovic, B. (2015) Global Income Distribution: From the Fall of the Berlin Wall to the Great Recession, World Bank Economic Review

Reviewer #2 (Remarks to the Author):

Review of “Poverty eradication in a carbon constrained world”

The paper uses a global consumption survey to estimate the carbon emission of various income groups in the world and the impacts of reducing poverty on global emissions. They test two scenarios, one in which extreme poverty is eradicated (the goal of the World Bank and international community) and one in which all people are given the income of the global middle class. They find that eradicating extreme poverty does not threaten the objective of stabilizing climate change below 2C. They also find that bringing everybody to the level of the global middle class would threaten this objective, with current technologies. They discuss the implication for achieving the SDG1 and the objective of the Paris Agreement.

This is a very important paper, with innovative, relevant, and solid results. There is no doubt for me that the paper should be published.

In particular, it complements very usefully recent World Bank reports (Hallegatte et al. 2016; Fay et al. 2015) that make the case that climate mitigation policies can be introduced without slowing down poverty reduction, but do not calculate the impact of doing so on global emissions. And it adds to previous estimates that were not based on consumption surveys and were therefore unable to account for intra-country heterogeneity. I think the paper should have a lot of influence on the debate on climate change and development.

There is one thing that needs to be corrected before the paper is published, however. It relates to how your technological assumptions are presented, and to your conclusion regarding technological fixes.

** Technology. Technology assumptions play a major role in the paper, and this needs to be clarified in a few places:

- Line 55: “growth in energy use is strongly coupled to economic growth and poverty reduction”, I think you mean correlated, and this is due to the technology that are used, and the consumption patterns (see (van Benthem 2015)). With different technologies and consumption patterns, this could change, so I would replace “coupled” by “correlated” and add that this relationship is not a law of physics that be remain valid over time: it can change, and it has changed in the past. See Box 1 in (Kalra et al. 2014) that shows how the energy intensity of GDP growth changed after the oil shock in 1973.

RE: we fully agree and use ‘correlated’ instead. The relevant part of the sentence now reads: “...growth in energy use is correlated with economic growth and poverty reduction , although the strength of this link is subject to change dependent on technology and consumption patterns (Steckel, Brecha et al. 2013).

- Line 109-112: “This assumes that additional production [...] is met with current capacity”: this is impossible and new capacities will have to be added. You want to say that this assumes that additional production is produced with similar technologies and similar energy intensity and similar carbon intensities. Line 112 you say that this leads to “conservative estimates” but the meaning is unclear: do you mean that you underestimate or overestimate emissions? I think you overestimate emissions because future technologies are likely to be more efficient than current one (see how a growing fraction of energy production comes from renewable in developing countries, and the empirical evidence in (van Benthem 2015)).

RE: you are correct. We have now added the additional required capacity (see response to previous reviewer) but do not adjust for more efficient technologies. While it is true that our estimations would be an ‘overestimation’, we don’t feel that this is the right term to use as we want to know explicitly the additional carbon emissions required ceteris paribus and these numbers provide a target for addition efficiency gains needed to compensate for the additional

carbon associated with poverty alleviation. The International Energy Outlook 2016 has provided a recent forecast for up to 2040 predicting small reductions in energy intensity and carbon intensity over the 2012-2040 period. EIA reports that these forecasts are highly uncertain as dependent on policies and regulations that will be implemented, as well as the potential role of new technologies. EIA data (Source: CO2 Emissions from Fuel Combustion (2016 Edition), OECD/IEA, Paris.) shows a declining ratio of CO2 emissions to real GDP until about 2000, then fairly flat thereafter. Since 2000 emissions per unit of energy use have, in fact, been rising, because of the greater use of coal. Because of the above considerations and the fact that we are adding the additional carbon up to 2030, not too far into the future, we believe assuming constant technology until 2030 is not too problematic.

Additional reference:

International Energy Outlook 2016 With Projections to 2040 DOE/EIA-0484(2016), May 2016 CO2 Emissions from Fuel Combustion (2016 Edition), OECD/IEA, Paris

- A good complement to your approach would be to look at how technologies need to change to achieve different poverty reduction goals without threatening climate objectives (this is what is done in (Rozenberg et al. 2015) with economic growth).

RE: this is now being done (see earlier response)

Overall, I think the fact that current technologies do not allow to bring everybody to the level of the middle class without threatening the climate objective does not mean that it is impossible to do so – it only means that technologies need to improve (which we know already). So your first message (even with current technology, the eradication of extreme poverty is not an issue) is much more robust than the second one (it's more difficult to bring people to middle class level).

RE: We agree with this reasoning and have adjusted the text to reflect this. The relevant section in the abstract now reads: “we find that eradicating extreme poverty does not jeopardize the climate target even in the absence of climate policies and with current technologies. On the other hand, bringing everybody to an income level of what can be considered the global middle class (between \$2.97 and \$ 8.44 PPP per day, i.e. between the 50th and 75th percentile), which is still by the standards of industrialized countries a fairly modest lifestyle, would have long-term consequences on achieving emission targets and would require much more carbon- efficient technologies than what we have now.”

I think that the abstract and text is misleading when it suggests that bringing people to middle class level would be a problem for emissions – you have to clarify that this is with current technologies and lifestyles. What you show, therefore, is that policies are needed. (In contrast, you show that extreme poverty can be eradicated without big impact on emission, even in the absence of climate policies.)

RE: We agree and adjusted the text accordingly (see previous response).

And I think that the conclusion that your results do not show that a “focus beyond technical fixes” is needed. To say that you would have to replace the energy and carbon intensity used in the projections by estimates of optimistic future intensities to show that technologies do not have enough potential. You are not doing this analysis and therefore you cannot conclude on the fact that technology is not enough. (Note that I’m not saying that this conclusion is wrong – only that your analysis does not support it.) This needs to be corrected in the abstract and in the text.

RE: There are numerous studies showing that technological advances have not been as successful as initially thought. As an example, most IPCC scenarios that have a good chance to keep the temperature below 2 degrees require negative emissions, some requiring widespread implementation of these technologies such as bio-energy production with CCS (BECCS), as early as 2030 (IIASA scenario database). However, implementing energy production combined with CCS is proving much harder than expected (Reiner, 2016). According to the Global CCS Institute (2016) there are 15 operational CCS projects around the world, capable of capturing about to 30 million tons per year of CO₂. While you are correct that we do not systematically compare the additional gain in carbon efficiency to compensate for the additional carbon, we do know that so far technology has not been able to keep up with additional emissions and our scenarios would require even more technological progress on top of what we would have otherwise. We amended the text to include this additional information.

Added Refs:

Reiner, DM (2016). “Learning through a Portfolio of Carbon Capture and Storage Demonstration Projects.” Nature Energy 1, 15011.

Global CCS Institute (2016). The Global Status of CCS: 2016, Summary Report. Melbourne, Australia.

**** Poverty line.** The paper makes multiple references to the \$1.25 poverty line, which is based on 2005 PPP exchange rate. Only a note in the figure mentions the change to the \$1.90 line with the 2011 PPP. It would be good to clarify the paper (which line is used and why). The two lines are designed to be equivalent so it is not a problem, but the exposition needs to be clarified. (And the abstract should not say that extreme poverty is currently represented by the \$1.25 line – it is represented by the \$1.90 line).

RE: We updated the poverty line from \$1.25 to \$1.9 according to the latest World Bank report and updated all the data and modeling results accordingly.

**** Studies of distributional impacts of fossil fuel subsidies.** You are right that few studies have been published on the distributional impacts of climate policies (even though you are missing a few, including (Rao 2013; Speck 1999; Metcalf 2008; Flues and Thomas 2015) and the discussion in (Fay et al. 2015). But a lot has been written on fossil fuel subsidies and their distributional impacts: similar arguments were made on these subsidies (removing them would increase poverty) but analysis of household surveys showed that removing them can reduce

poverty is part of the savings are used for poverty reduction actions (see reviews in Fay et al. 2015; Ruggeri Laderchi 2014; IMF 2013; Arze del Granado, Coady, and Gillingham 2012).

RE: Yes, you are right that in general there are quite a few studies that look at distributional effects of climate or energy policies but we kept our review mainly at the global level and there are much fewer relevant studies. Nevertheless, we appreciate the literature recommendation and have updated our literature review.

** Population scenarios. While I'm comfortable with the assumptions that the authors need to do to cope with imperfect data, I think they should explain better the change in population. If I understand well, country-level population changes are applied for income categories within countries, and there is no economic growth beyond the change in income linked to bringing people out of poverty. For a scenario, it looks really strange. What should be done is to run a scenario with population change and economic growth, and to take all the poor people in this scenario and to make them non-poor. That's complicated because there are only a few scenario for future poverty (the World Bank has the scenarios used in the Shock Waves report on the link between climate change and poverty, but they only go to 2030). One option for the authors is to explain that they focus on the incremental emissions from poverty reduction, compared with a scenario with no poverty reduction (basically, assuming that the RCP2.6 scenarios they look at does not reduce poverty at all – and compare with the same scenario with poverty reduction). I recommend they clarify this part.

RE: As referee mentioned, it is difficult to project poverty reduction within countries without reliable data on future economic growth and this is also not our intention in this paper. As we stated in the early part of this paper, we aim to show the additional carbon associated with lifting people out of poverty by 2030 according to this particular UN Sustainable Development Goal. Our population scenario is additional to the IPCC scenario. In this study, we assume that all people in extreme poverty will be moved out of poverty by 2030. With consideration of natural population growth (assuming growth rate is equal to low income country's average growth rate), we calculated how many people on average need to be moved out of poverty each year to be able to achieve the UN sustainable development goal (remove extreme poverty by 2030) using total population in extreme poverty divided by the number of years to 2030. For this paper, we do not want to make any assumptions of how this achieved or that it is dependent on any particular economic growth scenario (and assumptions about trickle down effects) but simply ask the question for the required additional carbon associated with this goal of poverty alleviation.

** Costs. Line 188-190: "Other more proven technologies [...] not be affordable by poorer countries." I think this statement is too strong. The world failed to provide electricity to Africa even with massive investments and using cheap fossil fuels, because of issues related to grid development and energy system management. The (small and decreasing) additional cost of renewable may well be more than compensated by how it makes it possible to provide decentralized energy, mitigating the issues related to system management and development of a national grid in low density countries. This statement should be changed.

RE: we agree. We have removed this statement

**** Effect on deforestation and other emissions. Line 376-378:** I do not really understand this point, and why the authors need it. But eradicating poverty would change livelihood, and hopefully will reduce extraction from ecosystems and therefore deforestation, so that the argument does not really work. I think the authors have to acknowledge that they cannot capture these effects. Deforestation hopefully would go in the right direction (less deforestation with higher income), so it should be okay. The authors also miss other aspects such as meat consumption, which could have large impacts.

RE: We do acknowledge that we cannot reasonably address this complex question in this paper. It is really beyond the scope of this paper as the literature is quite inconclusive on this topic. Most of the studies that link land use to consumption patterns are at the regional scale (Hubacek et al. 2009) or at best at the national level (DeFries and Pandey, 2010; Hubacek and Sun, 2001; Jonas et al. 2013), and frequently don't explicitly account for differences in income. The few global level studies show that land consumption seem to increase with higher income (Weinzettel et al. 2013; Yu et al. 2013). But at the lowest income categories the direction of impact is less clear-cut as we try to show with this paragraph and associated references. See also response to referee 3.

Meat consumption is included although not in the best possible way as meat consumption is subsumed as part of consumption for agricultural products. Higher income people would pay higher prices for agricultural products and thus cause higher carbon emissions. In this sense we have captured increases in meat consumption. A better way would have been to have a more disaggregated global model to better capture different types of food consumption. Part of the problem is the trade-off of sectoral detail and country coverage of different global multi-regional input-output models. We have added a note in the new subsection on limitations in the methods section.

Additional References:

DeFries, R. and D. Pandey (2010). "Urbanization, the energy ladder and forest transitions in India's emerging economy." Land Use Policy 27(2): 130-138.

Jonas, K., P. P. Glen and M. A. Robbie (2013). "Attribution of CO 2 emissions from Brazilian deforestation to consumers between 1990 and 2010." Environmental Research Letters 8(2): 024005.

Hubacek, Klaus and Laixiang Sun (2001). "A Scenario Analysis of China's Land Use Change: Incorporating Biophysical Information into Input-Output Modeling." Structural Change and Economic Dynamics Vol. 12/4, pp. 367-397

Hubacek, Klaus, Dabo Guan, John Barrett, and Thomas Wiedmann (2009). "Environmental implications of urbanization and lifestyle change in China: Ecological and Water Footprints". Journal for Cleaner Production. Vol. 17. pp. 1241-1248.

Weinzettel, Jan, Edgar G. Hertwich, Glen P. Peters, Kjartan Steen-Olsen, Alessandro Galli, Affluence drives the global displacement of land use, Global Environmental Change, Volume 23,

Issue 2, April 2013, Pages 433-438

Yu, Y., Hubacek, K., Feng, K., (2013). Tele-connecting local consumption to global land use. Global Environmental Change. Volume 23, Issue 5, Pages 1178–1186.

**** Other things**

Line 228: “Millenium Sustainability Goal” is a nice mix of MDG and SDG!

RE: we changed to SDG

Line 339: Sentence is not complete.

RE: The citation completes the sentence.

References

Arze del Granado, Francisco Javier, David Coady, and Robert Gillingham. 2012. “The Unequal Benefits of Fuel Subsidies: A Review of Evidence for Developing Countries.” *World Development* 40 (11): 2234–48. doi:10.1016/j.worlddev.2012.05.005.

Fay, Marianne, Stephane Hallegatte, Adrien Vogt-Schilb, Julie Rozenberg, Ulf Narloch, and Tom Kerr. 2015. “Decarbonizing Development : Three Steps to a Zero-Carbon Future.” Washington, DC: World Bank. <https://openknowledge.worldbank.org/handle/10986/21842>.

Flues, Florens, and Alastair Thomas. 2015. “The Distributional Effects of Energy Taxes.” *OECD Taxation Working Papers*. Paris: Organisation for Economic Co-operation and Development. <http://www.oecd-ilibrary.org/content/workingpaper/5js1qwkqrbv-en>.

Hallegatte, Stephane, Mook Bangalore, Laura Bonzanigo, Marianne Fay, Tamaro Kane, Ulf Narloch, Julie Rozenberg, David Treguer, and Adrien Vogt-Schilb. 2016. “Shock Waves: Managing the Impacts of Climate Change on Poverty.” *Climate Change and Development Series*. Washington, DC: World Bank.

IMF. 2013. *Energy Subsidy Reform – Lessons and Implications*. International Monetary Fund.

Kalra, Nidhi, Stephane Hallegatte, Robert Lempert, Casey Brown, Adrian Fozzard, Stuart Gill, and Ankur Shah. 2014. *Agreeing on Robust Decisions: New Processes for Decision Making under Deep Uncertainty*. Policy Research Working Papers. The World Bank. <http://elibrary.worldbank.org/doi/abs/10.1596/1813-9450-6906>.

Metcalf, Gilbert E. 2008. “Designing a Carbon Tax to Reduce US Greenhouse Gas Emissions.” *Review of Environmental Economics and Policy*, ren015.

Rao, Narasimha D. 2013. “Distributional Impacts of Climate Change Mitigation in Indian

Electricity: The Influence of Governance.” *Energy Policy* 61 (October): 1344–56.
doi:10.1016/j.enpol.2013.05.103.

Rozenberg, Julie, Steven J Davis, Ulf Narloch, and Stephane Hallegatte. 2015. “Climate Constraints on the Carbon Intensity of Economic Growth.” *Environmental Research Letters* 10 (9): 95006. doi:10.1088/1748-9326/10/9/095006.

Ruggeri Laderchi, Caterina. 2014. “Transitional Policies to Assist the Poor While Phasing out Inefficient Fossil Fuel Subsidies That Encourage Wasteful Consumption.” Contribution by the World Bank to G20 Finance Ministers and Central Bank Governors. <http://www.worldbank.org/content/dam/Worldbank/document/Climate/Transitional-Policies-Assist-Poor-Phasing-Out-Inefficient-Fossil-Fuel-Subsidies.pdf>.

Speck, Stefan. 1999. “Energy and Carbon Taxes and Their Distributional Implications.” *Energy Policy* 27 (11): 659–67. doi:10.1016/S0301-4215(99)00059-2.

van Benthem, Arthur. 2015. “Energy Leapfrogging.” *Journal of the Association of Environmental and Resource Economists* 2 (1): 93–132.

RE: thanks for the very helpful reference list. We have added a few to complement our literature review.

Reviewer #3 (Remarks to the Author):

I reviewed an earlier version of this article. The article has been revised substantially to include requested details on methodology and other suggestions. However, the analysis and its presentation remain unchanged, leaving many of the previously raised issues concerning troubling assumptions unaddressed. The conclusions at a qualitative level are relatively predictable, and their quantitative implications not reliable due to these assumptions. Nevertheless, given the importance of this question and that this is the first (known) attempt to provide an answer with a fair amount of work done, I would suggest the editor consider a revision subject to two important conditions: the authors include a new subsection on analysis caveats/assumption that brings together the problematic assumptions all together, their implication (direction of impact) for the conclusion for each assumption and if possible the overall estimates, and sensitivities where possible (as indicated below). Second, the translation of economic activity to climate sensitivity is seriously flawed (as described later) – the authors should stick to the metric of GHG emissions, drawing implications for aggregate impacts based on relative footprints of different income groups.

Assumptions in calculating GHG emissions:

- The authors in their baseline use current emissions intensities (from EORA) for the rest of the century, ignoring well-established trends of reducing energy intensity and climate policy. This assumption would probably overestimate the poor's future emissions.

RE: We do not intend to investigate different technology assumptions as this is done by the integrated assessment models. We 'only' establish the additional carbon emissions associated with poverty alleviation as a basis for calculating the reduction of annual carbon reduction required. We explicitly model the poverty alleviation for 2030, i.e. another 14 years, which is in line with the SDG. The assumption of constant carbon intensity is in line with available forecasts (and their difficulty given policy uncertainty) and previous trends. For example, The International Energy Outlook 2016 has forecast up to 2040 predicts small reductions in energy intensity and carbon intensity over the 2012-2040 period. EIA reports that these forecasts are highly uncertain as dependent on policies and regulations that will be implemented, as well as the potential role of new technologies. EIA data shows a declining ratio of CO2 emissions to real GDP until about 2000, slowing down afterwards globally and remaining flat for non-OECD and non Annex I parties.

To assume more or less constant carbon intensity is in line with estimates of the IEA outlook. We then add the additional carbon emissions caused by poverty alleviation to the carbon emissions associated with staying within 2 degrees increase and calculate the additional cumulative carbon emissions and increase in temperature if nothing else changes. We then calculate the carbon reduction required and ask whether this is feasible given current trajectories and technological progress and current technologies in the pipeline. We raise this last issue without investigating it further as this would be beyond the scope of the paper. We feel that the term 'overestimation' of emissions does not capture the intent of this thought experiment as we explicitly want to know the additional emissions ceteris paribus and then show what is needed in terms of carbon reduction.

- The authors assume the consumption pattern of the low-income segment would remain the same throughout the century – technology stays constant. This is also unlikely, though it is unclear in which direction the inaccuracy lies.

RE: in terms of technology assumption – see previous response. In terms of consumption patterns: you are correct, we do not know how they might change but we do know that for very low income groups they would stay fairly stable as low income folks spend large shares of their income for basic necessities such as food, shelter and clothing. We looked into the variation of consumption patterns of the lowest income groups in many poor countries and people with less than respectively \$1.90PPP and \$2.97PPP expenditure per day just don't show much variation and this is probably true not just across countries but even more so for each country across time. We are not aware of any study that provides us with likely consumption bundles over alternative ones. Moreover, the assumption of constant consumption is only made until 2030, and after 2030 we keep people in the higher income bracket to compute the temperature implication by 2100 (see also the earlier response about technology assumption).

- The analysis ignores capital investment and government expenditure, though the authors are transparent about this. However, why not make an assumption about the footprint of both categories based on historical data, and propagate that into the future? This would be entirely consistent with the nature of the analysis, and provide a more complete assessment of footprint.

RE: In the revised version, we include emissions associated with capital investment and government expenditure. Please see our response to the first referee.

- Authors ignore land-use based emissions, about which they are also transparent. However, they can provide more qualitative insight on the quantitative implications going forward, if there is no basis to do sensitivities.

RE: This is quite a complex issue and we do not think that we can do it justice in this paper. The literature on how poverty alleviation would change land use at a global level is quite inconclusive and really beyond what we could hope to achieve in this paper. Most of the studies that link land use to consumption patterns are at the regional scale (Hubacek et al. 2009) or at best at the national level (DeFries and Pandey, 2010; Hubacek and Sun, 2001; Jonas et al. 2013), and frequently don't explicitly account for differences in income. The few global level studies show that land consumption seem to increase with higher income (Weinzettel et al. 2013; Yu et al. 2013). See also response to referee 1.

Additional References:

DeFries, R. and D. Pandey (2010). "Urbanization, the energy ladder and forest transitions in India's emerging economy." Land Use Policy 27(2): 130-138.

Jonas, K., P. P. Glen and M. A. Robbie (2013). "Attribution of CO 2 emissions from Brazilian deforestation to consumers between 1990 and 2010." Environmental Research Letters 8(2): 024005.

Hubacek, Klaus and Laixiang Sun (2001). "A Scenario Analysis of China's Land Use Change: Incorporating Biophysical Information into Input-Output Modeling." Structural Change and Economic Dynamics Vol. 12/4, pp. 367-397

Hubacek, Klaus, Dabo Guan, John Barrett, and Thomas Wiedmann (2009). "Environmental implications of urbanization and lifestyle change in China: Ecological and Water Footprints". Journal for Cleaner Production. Vol. 17. pp. 1241-1248.

Weinzettel, Jan, Edgar G. Hertwich, Glen P. Peters, Kjartan Steen-Olsen, Alessandro Galli, Affluence drives the global displacement of land use, Global Environmental Change, Volume 23, Issue 2, April 2013, Pages 433-438

Yu, Y., Hubacek, K., Feng, K., (2013). Tele-connecting local consumption to global land use. Global Environmental Change. Volume 23, Issue 5, Pages 1178–1186.

- The authors use the average consumption pattern and population growth of low-income countries as a proxy for those of low-income people, including those in higher income countries. This is likely to be an overestimate. Low income countries also have elites with high shares of

total consumption – these countries’ average expenditure patterns likely do not resemble those of the poor in wealthier countries.

RE: We calculated the carbon footprint for different income groups in developing and developed countries separately based on different consumer expenditure survey databases. For developing countries, we use the World Bank consumption database, while for developed countries we collected consumption data from various sources including Eurostat, US BEA, and Japanese statistical office. Therefore, we did not use country averages to represent consumption pattern of the poor or the elites but used the consumption patterns by each countries lowest income group to represent lifestyles of the poor and the consumption patterns of the highest available income group to represent the elites. Within each income group we have to use averages and thus the variation within income groups of a specific country is not captured which is more of a problem when interested in capturing the top income folks (the so-called top 1 or 0.1% within a country) also as response rates of higher income earners tends to be lower in the World Bank database. (see also our response to referee 1).

Flaws in calculation of temperature impact:

- The authors add their emissions to an RCP2.6 trajectory of GHG concentrations, which make little sense. A socioeconomic pathway that yielded a 2.6-type emissions pathway would require substantial climate policy with transformative shifts in technology. This is entirely incompatible with a snapshot footprint of the poor from today. In an RCP2.6 world, the poor’s emissions may be entirely negligible. If at all, authors should use a BAU type RCP trajectory, and carefully interpret the result as a sensitivity from an increase in emissions akin to the level of emissions emitted from poverty eradication. Alternatively, and more accurately, the authors should estimate the aggregate emissions impacts (only) of relative differences in the per capita footprint composition for income groups, which is really the meat of the contribution here. In that vein, authors should in any case show direct and indirect footprints by income groups in some form.

RE:

With regards to the RCP discussion: we agree with the reviewer that our main contribution are the aggregate emissions impacts of poverty reduction. Contrasting our values against available carbon budgets for 2 degrees would be one reasonable way to present our results. Our choice of using the RCP2.6 trajectory is driven by the need to assess the consistency of the SDGs (to be achieved by 2030) with climate change impact targets usually expressed in term of keeping global temperature below 2 degrees above pre-industrial levels by 2100. Increases in temperature depend largely on the cumulative CO2 emissions released through to the end of this century. The standardized RCP scenarios were developed to provide stylized future developments of emissions that are as independent as possible from specific socio-economic and technological assumptions capable of supporting the parallel development of climate models and new socio-economic scenarios (Moss et al., 2010). More detailed scenarios that are consistent with different RCPs are produced by IAM models (see, IIASA database and references therein).

We use the RCP2.6 to provide the cumulative emission pathway consistent with the climate objective of limiting warming below 2 degrees, so that we can focus on the added impact of eradicating poverty. The reason not to use RCP8.5, the so-called business as usual 'baseline' scenario (that, e.g., assumes no technological change and energy intensity improvements) is that it does not have any target mitigation policy and is thus not helpful to put our result in the climate debate context.

Following your suggestion, we have included direct and indirect emissions in figure 1as well as added some text about the carbon intensity per dollar spent for each expenditure group.

Additional reference:

Moss et al. (2010). "The next generation of scenarios for climate change research and assessment" Nature 463, 747-756 (11 February 2010) | doi:10.1038/nature08823

Smaller points:

- the authors confuse wealth for consumption towards the end of the article

RE: Thanks for pointing this out. We changed to income.

- the added paragraph on multidimensional poverty is welcome, but the statement about income being a good proxy for the poor is incorrect. Better to leave the discussion as an open caveat.

RE: We would agree with income not being a good proxy. Our sentence was as follows: "Despite limitations, quantifying the 'climate-development conflict' through greenhouse gas emissions associated with energy consumption (Rao, Riahi et al. 2014) is still a very useful first approximation as energy is part of many household consumption activities as well as being an essential input to production of goods and services in all stages of global supply chains." We don't just use income but the associated consumption bundle as represented by a range of consumption items as elicited through expenditure surveys and the energy that is directly and indirectly consumed. Thus we go beyond just using a single number (i.e. income) to characterize poverty.

Additional references:

Rao, N. D., et al. (2014). "Climate impacts of poverty eradication." Nature Clim. Change 4(9): 749-751.

Steckel, J. C., et al. (2013). "Development without energy? Assessing future scenarios of energy consumption in developing countries." Ecological Economics 90: 53-67.

Reviewers' comments:

Reviewer #1 (Remarks to the Author):

The authors provided several improvements to the manuscript, taking into account important concerns raised by the three reviewers. I still have two major comments for this second round of revision:

The bridge matrix (i.e. matrix of coefficient used to map consumption sectors to production sectors) that is now attached as a supplementary information to the paper, is a global bridge matrix - in the sense that there is no country specific matrix and looks all but arbitrary. The coefficients of this matrix are however likely to change across countries and not constant across consumption sectors. In fact, at this stage, the work is essentially based on arbitrary choices made by the authors. This is standard in the field but is clearly a limitation of this kind of work. A range of bridge matrices should rather be used, with probabilistic variations of the coefficients of the bridge matrix. This would seriously reduce the arbitrary feeling we get and increase the confidence we would get in final estimates.

Second, I am still not convinced by a very strong assumption made by the authors on the evolution of the consumption bundles of the poorest segments of the population (that is: consumption bundles are kept constant by 2030). The authors write that there is no work enabling them to make alternative assumptions. This overlooks an entire field of literature on the evolution of consumption patterns over time and class after T. Veblen's work (1898). A prospective work of the sort proposed by the author should present different possible scenarii, rather than keeping the consumption bundle of lower income groups constant. Otherwise, we can still worry about the robustness of the results.

In brief: I personally believe some extra modifications would be needed in order to give this paper all the originality and robustness it could (and should) have for a publication in Nature Communications- this would still require some important evolution however.

Reference:

T. Veblen (1898). Theory of the Leisure Class.

Reviewer #2 (Remarks to the Author):

Thanks to the authors for the responses and revision of their paper. I still think that this is a really important piece of work – with the potential to influence many discussions. So I repeat my assessment that this paper should be published.

I still think that the paper can be improved, and I suggest here a few ideas to do so.

First, in spite of an improvement in the text, the use of scenarios is still a bit confusing. It's due in part to the use of two approaches: one is based on the RCP2.6; the other based on an idealized mitigation pathway with constant rate of emission reduction. I have to say that the second approach is much more convincing and simple to explain, but is barely explained in the text (and from the main text I did not understand how it related to the RCP approach). I would take the text from the Supplementary material and use it in the main text:

"Average annual CO₂ emissions reduction rates for the period 2017-2100 330 corresponding to each poverty reduction goal required to stay below 2°C with 66% probability are also shown in the figure. Using a 331 standard simplified approach for the mitigation scenarios as, for example, in Jackson et al. (2015)⁵⁹, we find that the mitigation

rates 332 needed to stay within 2 degrees are -4.4%/yr if we assume constant 2016 emission rates (36.4Gt/year of CO₂, just 0.2% higher than 333 2015, Le Quéré et al. 201658) without poverty targets, -4.5%/yr if we add to the base the carbon needed to eliminate poverty 334 (<\$1.90/week), and -5.5%/year if we add to the baseline the carbon needed to bring people to the \$2.97-\$8.44 range of income per 335 week. Specifically, the average annual mitigation rate for the incremental carbon emissions from eliminating extreme poverty is 2.8% 336 higher (0.1 percentage point increase) than without poverty reduction goals. The mitigation rate for the additional carbon from 337 bringing people to \$2.97-\$8.44 per week range of income is 27.03% higher (1.1 percentage point increase) than without poverty 338 reduction goals (see Figure 5)."

I think this paragraph makes your point beautifully: compared to reference mitigation pathway, eradicating extreme poverty increases the effort by less than 3% - so it does not matter. Bringing everybody to the global middle class level is much more ambitious since it increases the effort by more than 25%. [By the way, I would put these numbers in the abstract.]

After you have made this point, you can add results in terms of temperature based on RCP2.6 but (1) explaining that this is a completely different exercise (note in particular that RCP2.6 has negative emissions so it would overshoot in your figure 5 with the carbon budget); (2) without saying that RCP2.6 is a baseline (this is confusing since baselines are no-mitigation scenario in most papers!).

Second, the discussion on technology (top of page 6) is both not necessary and not convincing. The penetration of solar and wind energy has exceeded even the most optimistic forecasts done in the last 30 years, and costs have decreased at least ten times faster than expected. Electricity storage is following the same path. Solar is now commercially available at 5 cents a kWh. I do not think you can say that "technological advances have not been as successful as initially thought". And this is not needed for your paper. You can just conclude on the fact that eradicating extreme poverty does not change the challenge, while bringing everybody to the global middle class level makes either the technological challenge much more difficult, or implies redistribution (to reduce emissions from the richest). I do not think a discussion of CCS is relevant in your conclusion.

Another point – which is really important I think – is that the RCP2.6 represents such a massive change in economic structure and patterns and technology. If you do not think innovation will deliver in the domain, then it makes little sense to use the RCP2.6 as your reference scenario!

Third, I'm not sure I understand how your approach to increase global population (within each income segment) manages to be consistent with the UN projection. If you use the projection for low-income countries for the corresponding income segment in each country, there is no reason that the total project for one given country is equal to what the UN projects. Take Malawi: maybe 70 percent of the population is in extreme poverty, and you will make this segment grow like the population of low-income countries (i.e. like Malawi). The remaining 30 percent will grow more slowly. How can the sum of these segmentation be consistent with UN's projection for Malawi? I may have missed something in the explanation.

Finally, an additional edit would be useful, to make the text more efficient. For instance, your introduction is very long compared with the space for methodology. Personally I would prefer less background (it's pretty straightforward) and more methodology in the document. There are also sentences that are not very clear (e.g. line 316 "We present un-abated emissions"). I think the contribution of your paper is important and it deserves to be presented as efficiently as possible.

Thanks for this contribution.

Reviewer #3 (Remarks to the Author):

The responses aren't entirely responsive to the revision requests. Regarding capital and govt expenditure, I/O accounts are just a snapshots of historical expenditure, and not an account of what expenditure is needed to support income growth. Further, scaling capex/govt expenditure in proportion to household income is likely in the opposite direction of what is required to bring people out of poverty. The authors should add a few sentences in the Limitations section on the arbitrariness of scaling capital formation and government expenditure, and the caveat that their assumption is not based on any understanding of actual investment requirements for poverty eradication.

With this addition, although I am uncomfortable with the number of simplifications in this analysis, I feel the article will have a level of transparency that will allow careful readers to judge the value of the analysis themselves. Overall, the conclusions should be understood as qualitative, rather than quantitative. If this can be made explicit in the written text, the article would enhance its credibility.

Two minor points:

The authors unduly defend a constant future carbon intensity in their results - it wouldn't hurt (rather, it would help their case) to acknowledge that carbon intensity of poverty eradication may be less with climate policies (e.g. INDCs), which would only strengthen their conclusion.

Authors ignore non-commercial biomass, which is why the direct emissions for the poorest group are so low. While this is apparent from the limitations section, it wouldn't hurt to state this in the main text.

Reviewers' comments:

Reviewer #1 (Remarks to the Author):

The authors provided several improvements to the manuscript, taking into account important concerns raised by the three reviewers. I still have two major comments for this second round of revision:

The bridge matrix (i.e. matrix of coefficient used to map consumption sectors to production sectors) that is now attached as a supplementary information to the paper, is a global bridge matrix - in the sense that there is no country specific matrix and looks all but arbitrary. The coefficients of this matrix are however likely to change across countries and not constant across consumption sectors. In fact, at this stage, the work is essentially based on arbitrary choices made by the authors. This is standard in the field but is clearly a limitation of this kind of work. A range of bridge matrices should rather be used, with probabilistic variations of the coefficients of the bridge matrix. This would seriously reduce the arbitrary feeling we get and increase the confidence we would get in final estimates.

RE: The World Bank's Consumer Expenditure Survey (CES) Product (global consumption database <http://datatopics.worldbank.org/consumption/detail>) and the global multi-regional input-output database (<http://worldmrio.com/simplified/>) have been developed to enable global analyses and are uniform across countries. There are a number of options on how to assign consumer consumption items to the Eora production sectors, which we explicitly consider using uncertainty analysis. For most sectors the allocation is straightforward and allows only one possible match. For example, in the World Bank global consumption database, 'Processed fish and seafood', 'Cheese, butter and margarine', 'Other edible oil and fats' can only fit to Eora sector Food & Beverages. However, there are some cases that one consumption item category may be produced from multiple economic sectors. For example, some food consumption categories can either be sold directly from the farm and thus would be linked to Eora sector 'Agriculture' or have been processed and thus need to be linked to Eora sector 'Food & Beverages'. However, there is no good reference to split the aggregate consumption categories, such as 'Other Cereals, Flour and Other Products', 'Fresh or Chilled Vegetables Other than Potatoes', into 'Agriculture' and 'Food & Beverages' sectors'. And, as raised by the reviewer, the allocation of an aggregate consumption category might vary from country to country. Most studies ignore this problem (Fessau and Mattonetti, 2013; Fesseau and van de Ven, 2014), and just equally distribute the aggregate category in consumption (as e.g. done by the World Bank's Global Consumption Database) or when linking environmental accounts to input-output accounts (e.g. Lenzen 2011), or numerous input-output studies linking CES to input-output categories (e.g. (Kok, Benders et al. 2006, Ornetzeder, Hertwich et al. 2008, Weber and Matthews 2008, Druckman and Jackson 2009, Baiocchi, Minx et al. 2010; Steen-Olsen, 2016; Wiedenhofer, 2017). We followed this practice for this study but based on the criticism raised by the referee we have added an uncertainty analysis using an allocation approach based on different possible 'extreme' bridging matrices. We select the possible IO sectors that can be linked to the CES sectors. For the maximum value, we assign the CES category to the sector with

the highest emissions multiplier, for the lowest possible value, we assign the CES category to the sector with the lowest emissions multiplier. We then take these extreme or maximum deviations from the allocation we had chosen and calculate the deviation for each household category (see the method section in the manuscript) and show the range through error bars for each household category.

The uncertainty analysis for the shows that re-allocation of the consumption categories that may fall into multiple economic sectors has a relatively small impact on per capita footprints of different household groups (see figure below). The uncertainty for the carbon footprint (CF) for all household groups is in the range of less than 2% between the max and the min, and even less of an issue for our scenarios only involving the lowest 2 income categories. We show the sectors that potentially overlap with multiple IO sectors in the five bridge matrices in the supporting information. A sector match is indicated by 1 whereas a range of matches is represented by 0-1 indicating that share ranging somewhere between those values but we do not have additional information to assign a value. As developed countries are at the top of the global income distribution, they are not included in the poverty alleviation exercise.

Uncertainty analysis for per capita carbon footprint of five household groups

References:

Baiocchi, G., et al. (2010). "The Impact of social factors and consumer behavior on carbon dioxide emissions in the United Kingdom." Journal of Industrial Ecology 14(1): 50-72.

- Druckman, A. and T. Jackson (2009). "The carbon footprint of UK households 1990–2004: A socio-economically disaggregated, quasi-multi-regional input–output model." *Ecological Economics* 68(7): 2066-2077.
- Fessau, M. and M. L. Mattonetti (2013). *Distributional measures across household groups in a national accounts framework. Results from an experimental cross-country exercise on household income, consumption and saving. STD/CSTAT/WPNA(2013)10/RD. OECD. Paris*
- Fesseau, M., van de Ven, P., 2014. *Measuring inequality in income and consumption in a national accounts framework. OECD Statistics Brief 19, 1-12.*
- Kok, R., R. M. J. Benders and H. C. Moll (2006). "Measuring the environmental load of household consumption using some methods based on input–output energy analysis: A comparison of methods and a discussion of results." *Energy Policy* 34(17): 2744-2761.
- Lenzen, M., 2011. *Aggregation versus disaggregation in input-output analysis of the Environment. Economic Systems Research* 23, 73 - 89.
- Ornetzeder, M., et al. (2008). "The Environmental Effect of Car-free Housing: A Case in Vienna." *Ecological Economics* 65(3): 516-530.
- Steen-Olsen, K., Wood, R. and Hertwich, E. G. (2016), *The Carbon Footprint of Norwegian Household Consumption 1999–2012. Journal of Industrial Ecology*, 20: 582–592.
- Kok, R., et al. (2006). "Measuring the environmental load of household consumption using some methods based on input–output energy analysis: A comparison of methods and a discussion of results." *Energy Policy* 34(17): 2744-2761.
- Weber, C. L. and H. S. Matthews (2008). "Quantifying the global and distributional aspects of American household carbon footprint." *Ecological Economics* 66(2-3): 379-391.
- Wiedenhofer, D., Guan, D., Liu, Z., Meng, J., Zhang, N., Wei, Y.-M., 2017. *Unequal household carbon footprints in China. Nature Clim. Change* 7, 75-80.

Second, I am still not convinced by a very strong assumption made by the authors on the evolution of the consumption bundles of the poorest segments of the population (that is: consumption bundles are kept constant by 2030). The authors write that there is no work enabling them to make alternative assumptions. This overlooks an entire field of literature on the evolution of consumption patterns over time and class after T. Veblen's work (1898). A prospective work of the sort proposed by the author should present different possible scenarii, rather than keeping the consumption bundle of lower income groups constant. Otherwise, we can still worry about the robustness of the results.

In brief: I personally believe some extra modifications would be needed in order to give this paper all the originality and robustness it could (and should) have for a publication in *Nature Communications*- this would still require some important evolution however.

Reference:

T. Veblen (1898). *Theory of the Leisure Class.*

RE: We agree that there is a huge amount of literature focusing on consumption patterns within income/expenditure categories although less so at the global level as discussed in this paper; but the goal of the paper is not to predict carbon emissions for the year 2030 or 2100 but to investigate 2 poverty alleviation scenarios (of moving people from a lower category to a higher one) by 2030 using available detailed data on consumption for different expenditure and income categories from the World Bank and other statistical agencies. We want to investigate these counterfactual scenarios based on detailed available data and do not want to dilute these scenarios by modifying existing consumption bundles in all income categories other than moving people from one category and the associated consumption bundle to the next. We re-emphasize this focus and the potential limitation at various places in the paper. We do not think that this is an unreasonable assumption to make that the composition of consumption is going to stay fairly constant for the next 12.5 years or so (i.e. until 2030) within the poorest groups in the poorest countries of the global economy. This is due to the fact that poor people are spending most of their income on food, clothing and shelter, which is also shown in the table below. The table shows the average and standard deviation for major expenditure categories showing that for the extremely poor (less than \$1.9) and the less than \$2.97 category the main consumption items account for about three quarters of the budget and that is across the 90 poorest countries. This shows that there is quite a bit of stability in terms of expenditure patterns across the poor in that three quarters of the expenditure is for food, shelter and housing, and the remaining expenditure categories only make small contributions to the expenditure. This is also reflected in the literature. The most important determinant of the carbon footprint is income (see e.g. Wier et al. 2001, Weber and Matthews, 2008; Minx et al. 2013; Ahmad et al. 2015), which we consider explicitly in our two poverty alleviation scenarios but other determinants of per capita carbon footprints such as urban characteristics (Kennedy, 2009), population density (Jones and Kammen, 2014), lifestyles (Baiocchi, 2010), household size (Jones, et al 2011), etc are not considered in this study as these do not exist at the global level and most of the countries we include in our study, and are only available for specific countries or selected cities. In summary, we feel quite comfortable with the ‘constant consumption assumption’ of the poor given that 1) income and composition are the most important and we change them explicitly based on detailed available information; 2) we are only interested in the counterfactual poverty alleviation scenario and their carbon implications; and stress that multiple times all over the paper; 3) there is a lack of available studies for most countries we are considering; 4) relative stability of consumption patterns of the poorest in the poor countries; and 5) assumption is only relevant for 13 years, i.e. until 2030, given that the SD goals want to achieve the poverty alleviation goals considered here by 2030. This discussion of assumptions has been added to the limitation section.

Table: Average share and standard deviation of main consumption items for extreme poverty and low expenditure group

	Extreme Poverty		Low Expenditure	
	Average share of consumption	SD	Average share of consumption	SD
Food and Beverages	0.60	0.10	0.58	0.10
Clothing and Footwear	0.06	0.02	0.06	0.03
Housing	0.08	0.06	0.09	0.07

Additional References:

- Ahmad, S., Baiocchi, G. & Creutzig, F. *CO2 Emissions from Direct Energy Use of Urban Households in India. Environmental Science & Technology* **49**, 11312-11320, doi:10.1021/es505814g (2015).
- Weber, C. L. & Matthews, H. S. *Quantifying the global and distributional aspects of American household carbon footprint. Ecological Economics* **6**, 379-391 (2008).
- Wier, M., Lenzen, M., Munksgaard, J. & Smed, S. *Effects of Household Consumption Patterns on CO2 Requirements. Economic Systems Research* **13**, 259-274, doi:<http://dx.doi.org/10.1080/09537320120070149> (2001).
- Minx, J. et al. *Carbon footprints of cities and other human settlements in the Uk. Environmental Research Letters* **8**, 035039 (2013).
- Kennedy, C. et al. *Greenhouse Gas Emissions from Global Cities. Environmental Science & Technology* **43**, 7297-7302, doi:10.1021/es900213p (2009).
- Baiocchi, G., Minx, J. & Hubacek, K. *The Impact of Social Factors and Consumer Behavior on Carbon Dioxide Emissions in the United Kingdom. Journal of Industrial Ecology* **14**, 50-72, doi:10.1111/j.1530-9290.2009.00216.x (2010).
- Jones, C. & Kammen, D. M. *Spatial Distribution of U.S. Household Carbon Footprints Reveals Suburbanization Undermines Greenhouse Gas Benefits of Urban Population Density. Environmental Science & Technology* **48**, 895-902, doi:10.1021/es4034364 (2014).
- Jones, C. M. & Kammen, D. M. *Quantifying Carbon Footprint Reduction Opportunities for U.S. Households and Communities. Environmental Science & Technology* **45**, 4088-4095, doi:10.1021/es102221h (2011).

Reviewer #2 (Remarks to the Author):

Thanks to the authors for the responses and revision of their paper. I still think that this is a really important piece of work – with the potential to influence many discussions. So I repeat my assessment that this paper should be published.

I still think that the paper can be improved, and I suggest here a few ideas to do so.

First, in spite of an improvement in the text, the use of scenarios is still a bit confusing. It's due in part to the use of two approaches: one is based on the RCP2.6; the other based on an idealized mitigation pathway with constant rate of emission reduction. I have to say that the second approach is much more convincing and simple to explain, but is barely explained in the text (and from the main text I did not understand how it related to the RCP approach). I would take the text from the Supplementary material is use it in the main text:

"Average annual CO2 emissions reduction rates for the period 2017-2100 corresponding to each poverty reduction goal required to stay below 2°C with 66% probability are also shown in the figure. Using a standard simplified approach for the mitigation scenarios as, for example, in Jackson et al. (2015)59, we find that the mitigation rates needed to stay within 2 degrees are -4.4%/yr if we assume constant 2016 emission rates (36.4Gt/year of CO2, just 0.2% higher than 2015, Le Quéré et al. 201658) without poverty targets, -4.5%/yr if we add to the base the carbon needed to eliminate poverty 334 (<\$1.90/week), and -5.5%/year if we add to the baseline the carbon needed to bring people to the \$2.97-\$8.44 range of income per 335 week. Specifically, the average annual mitigation rate for the incremental carbon emissions from eliminating extreme poverty is 2.8% higher (0.1 percentage point increase) than without poverty reduction goals. The mitigation rate for the additional carbon from bringing people to \$2.97-\$8.44 per week range of income is 27.03% higher (1.1 percentage point increase) than without poverty reduction goals (see Figure 5)."

I think this paragraph makes your point beautifully: compared to reference mitigation pathway, eradicating extreme poverty increases the effort by less than 3% - so it does not matter. Bringing everybody to the global middle class level is much more ambitious since it increases the effort by more than 25%. [By the way, I would put these numbers in the abstract.]

After you have made this point, you can add results in terms of temperature based on RCP2.6 but (1) explaining that this is a completely different exercise (note in particular that RCP2.6 has negative emissions so it would overshoot in your figure 5 with the carbon budget); (2) without saying that RCP2.6 is a baseline (this is confusing since baselines are no-mitigation scenario in most papers!).

RE: we have followed both recommendations and added the required % reduction in the abstract and moved the discussion of additional required reductions in the main text.

But to clarify our approach in response to the use of RCP: The only scenario analysis we are performing is to calculate the carbon emissions that would occur if we move people out of poverty by 2030. We then add these additional emissions to the carbon emission pathway of the RCP2.6 (see also our response to your 3rd question below). Finally, we ask for the additional

reduction of carbon intensity required to compensate for this additional emissions from the poverty alleviation scenario.

Second, the discussion on technology (top of page 6) is both not necessary and not convincing. The penetration of solar and wind energy has exceeded even the most optimistic forecasts done in the last 30 years, and costs have decreased at least ten times faster than expected. Electricity storage is following the same path. Solar is now commercially available at 5 cents a kWh. I do not think you can say that “technological advances have not been as successful as initially thought”. And this is not needed for your paper. You can just conclude on the fact that eradicating extreme poverty does not change the challenge, while bringing everybody to the global middle class level makes either the technological challenge much more difficult, or implies redistribution (to reduce emissions from the richest). I do not think a discussion of CCS is relevant in your conclusion.

RE: While you are right that there have been huge advances in terms of renewables and their competitiveness, we stick with their contribution globally and in the country examples. Fact is that the carbon intensity has not declined at significant rates and we stick with predictions of the IEA for the next 13 years. We agree on the basic message and emphasize that the global middle class scenario makes an already difficult task so much more difficult. The discussion of CCS and BECCS, which are important parts of the IPCC scenario exercises that keep us within an increase of 2° C help make this point.

Another point – which is really important I think – is that the RCP2.6 represents such a massive change in economic structure and patterns and technology. If you do not think innovation will deliver in the domain, then it makes little sense to use the RCP2.6 as your reference scenario!

RE: we agree with the point raised by the reviewer, but would like to note that the use of RCP2.6 is just to assess the climate target abstracted from possible socio-economic scenarios that can produce them in agreement with (see e.g. Moss et al. 2010; van Vuuren et al. 2011). According to the IPCC AR5: “The RCPs ARE NOT [emphasize added] associated with unique socioeconomic assumptions or emissions scenarios but can result from different combinations of economic, technological, demographic, policy, and institutional futures” (Wayne, 2013, p. 8). “RCPs each describe an emission trajectory and concentration by the year 2100, and consequent forcing. Each trajectory represents a specific synthesis drawn from the published literature. From this ‘baseline’, researchers can then test various permutations of social, technical and economic circumstances.” (Wayne, 2013, p. 9) And this is exactly how we are using the RCP as a reference emission trajectory to assess the carbon consequences of poverty alleviation. We do not make any assumption of whether innovation will deliver or not. We ‘just’ assume, based on the IEA report, that in the next 12 years that might not be major changes in

carbon intensity. We only calculate the additional gains in reduction required to offset the additional carbon emissions from poverty alleviation.

Additional Reference:

IPCC, 2014: Climate Change 2014: Synthesis Report. Contribution of Working Groups I, II and III to the Fifth Assessment Report of the Intergovernmental Panel on Climate Change [Core Writing Team, R.K. Pachauri and L.A. Meyer (eds.)]. IPCC, Geneva, Switzerland, 151 pp.

Moss et.al. 2010. The next generation of scenarios for climate change research and assessment, Nature, doi:10.1038/nature08823

van Vuuren et al. (2011). "The representative concentration pathways: an overview. Climatic Change (2011) 109:5–31, DOI 10.1007/s10584-011-0148-z

Wayne, G. 2013. "The Beginner's Guide to Representative Concentration Pathways". Skeptical Science. https://skepticalscience.com/docs/RCP_Guide.pdf

Third, I'm not sure I understand how your approach to increase global population (within each income segment) manages to be consistent with the UN projection. If you use the projection for low-income countries for the corresponding income segment in each country, there is no reason that the total project for one given country is equal to what the UN projects. Take Malawi: maybe 70 percent of the population is in extreme poverty, and you will make this segment grow like the population of low-income countries (i.e. like Malawi). The remaining 30 percent will grow more slowly. How can the sum of these segmentation be consistent with UN's projection for Malawi? I may have missed something in the explanation.

Finally, an additional edit would be useful, to make the text more efficient. For instance, your introduction is very long compared with the space for methodology. Personally I would prefer less background (it's pretty straightforward) and more methodology in the document. There are also sentences that are not very clear (e.g. line 316 "We present un-abated emissions"). I think the contribution of your paper is important and it deserves to be presented as efficiently as possible.

Thanks for this contribution.

RE: It would be ideal to estimate population growth based on the population projections for each income group in each country until 2100. However, these data are not available. On the other hand, we understand the concern the reviewer raised that different countries may have different population growth rates and using the average growth rate for low income countries for the low consumption group may lead to relatively large uncertainty. Given the predictions available from the UN World Population Prospects (<https://esa.un.org/unpd/wpp/>) we have two choices: 1) to use predictions that represent income segments and thus requires averaging across countries (the average population growth rate for countries within an income bracket) based on estimates by the UN, for example, we use the average population growth rate of low

income countries to represent the population growth for the extreme poverty group and the consumption group of >\$2.97 per day, and use the population growth rate of lower middle income countries for the consumption group of \$2.97-\$8.44 per day; or 2) to use country specific predictions but ignore the changing composition of expenditure groups (population growth rate by country). Both estimates have shortcomings. Applying the national average growth rate to the low income group within a country may lead to an underestimation of population growth for low income groups as their growth rate might be higher than the national average (Price, 2013) However, using the average population growth rate across all low income countries may also lead to uncertainty because countries have different population growth rates even though they all fall into the low income country category. In this study, we use option 1) i.e. using the average growth rate of the respective country group mainly because the country specific predictions are not an option to us anymore after we introduce our poverty alleviation scenarios as we significantly would influence income and thus fertility and mortality rate of these low income countries and the country specific predictions by the UN would not be applicable to our countries any more. When comparing the two approaches, we find that option 2 provides 13% lower carbon emissions under scenario 1 and 18% lower under scenario 2 than by using option 1.

Additional Reference

Price, J., 2013, How income Affects Fertility, Institute of Family Studies, (<https://ifstudies.org/blog/how-income-affects-fertility>).

Reviewer #3 (Remarks to the Author):

The responses aren't entirely responsive to the revision requests. Regarding capital and govt expenditure, I/O accounts are just a snapshots of historical expenditure, and not an account of what expenditure is needed to support income growth. Further, scaling capex/govt expenditure in proportion to household income is likely in the opposite direction of what is required to bring people out of poverty. The authors should add a few sentences in the Limitations section on the arbitrariness of scaling capital formation and government expenditure, and the caveat that their assumption is not based on any understanding of actual investment requirements for poverty eradication.

RE: There are two separate types of capital/government expenditure that are potentially relevant to this study. One is accompanying income growth (e.g. the relevant infrastructure investments that are needed and accompanying increase in production to meet this final demand), which is the one we added based on comments/requests for referees in the previous round. The other type of investment is the one that leads to poverty alleviation and triggers income growth for poor households, which is the one reviewer emphasized. There are a number

of ways that government expenditure can lead to poverty alleviation such as direct payment transfer (cash transfer) which are being trialed or implemented in Brazil¹ and India², and direct and indirect subsidies (e.g. fuel subsidies, food subsidies, poverty programs)^{3,4}. However, we did not include any such payment transfer and/or subsidies or similar mechanism to increase the income of the poor as outlined in these scenarios due to the complexity of these wealth distribution mechanism. However, this problem is partially mitigated by the use of expenditures instead of actual income, which is hard to measure in poorer countries. There is evidence that households in poorer countries have negative savings as they appear to spend more than they earn (see, e.g., Fesseau and van de Ven 2014). The fact that these types of transfer are implicit in the expenditure data is the standard justification for choosing them over income in many studies.

We added a brief discussion to the limitation section.

The scaled investment and government expenditure is likely to have some benefits for the poor but it is hard to estimate which benefits they accrue. For example, one might argue that a lot of the infrastructure investment benefits more the middle class, who can extract greater benefit from such investments. A simple example is road-building which leads to increased vehicular mobility. Those in extreme poverty have little to benefit from this road-building. We thus see these expanded government expenditure and investment as accompanying the expansion in producing those additional goods and services associated with poverty alleviation but they are not the trigger of poverty alleviation. These initial transfer payments are not explicitly included in this study. We do not ask the question of how to alleviate poverty?

Additional References:

Fesseau, M. and P. van de Ven. 2014. *Measuring inequality in income and consumption in a national accounts framework*. OECD Statistics Brief(19): 1–12

The World Bank. *Bolsa Família: Changing the Lives of Millions in Brazil*. The World Bank, Washington, DC. <http://go.worldbank.org/3QI1C7B5U0>. (accessed 4/28/2017).

SEWA Bharat. *A Little More, How Much It Is...Piloting Basic Income Transfers in Madhya Pradesh, India*. SEWA Bharat. New Delhi, India. (2014).

Kojima, M. *Fossil fuel subsidy and pricing policies: recent developing country experience*. (World Bank Policy Research Working Paper 7531, 2016).

Hidrobo, M., Hoddinott, J., Peterman, A., Margolies, A. & Moreira, V. *Cash, food, or vouchers? Evidence from a randomized experiment in northern Ecuador*. *Journal of Development Economics* **107**, 144-156, doi: [10.1016/j.jdeveco.2013.11.009](https://doi.org/10.1016/j.jdeveco.2013.11.009) (2014).

With this addition, although I am uncomfortable with the number of simplifications in this analysis, I feel the article will have a level of transparency that will allow careful readers to judge the value of the analysis themselves. Overall, the conclusions should be understood as

qualitative, rather than quantitative. If this can be made explicit in the written text, the article would enhance its credibility.

RE: We have added further description to limitations and elsewhere in the document (see responses to other questions/comments).

Two minor points:

The authors unduly defend a constant future carbon intensity in their results - it wouldn't hurt (rather, it would help their case) to acknowledge that carbon intensity of poverty eradication may be less with climate policies (e.g. INDCs), which would only strengthen their conclusion.

RE: Our scenarios and assumptions are only until the year 2030. While it is true that carbon intensities might change by 2030 especially given the INDCs, we do not want to explicitly model the impacts these policies have but strictly ask what additional reductions would be required if we fulfil the poverty reduction target stated in the SDG. The INDCs have just being proposed and to which extent they will be achieved is rather uncertain and not topic of this paper. Given the recent forecasts in the International Energy Outlook 2016 which only predicts small reductions in energy intensity and carbon intensity over the 2012-2040 period and given that these forecasts are highly uncertain and dependent on policies and regulations that will be implemented we feel relatively comfortable to stick with constant carbon intensity for the next decade or so. We have made this assumption and the justification in the main text and the suggested limitation section.

Authors ignore non-commercial biomass, which is why the direct emissions for the poorest group are so low. While this is apparent from the limitations section, it wouldn't hurt to state this in the main text.

RE: We have also mentioned this now in the main text.

REVIEWERS' COMMENTS:

Reviewer #1 (Remarks to the Author):

I would like to thank the authors for providing an argued and referenced response. I am however not convinced by the arguments brought forward to defend the choices made relative to the (non)evolution of consumer baskets over time.

The authors stress that income is the key determinant of carbon footprints - this is true today. But isn't the key question of interest here precisely to what extent can this change - under behavioural changes, technological changes and public policies- to enable poverty eradication and carbon mitigation? This is a crucial point and is convincingly addressed by the authors. I would still not recommend publication at this stage.

Reviewer #2 (Remarks to the Author):

Thanks for this revised version, which answers my comments.

Reviewer #3 (Remarks to the Author):

No comments to authors